# SpecDec++: Boosting Speculative Decoding via Adaptive Candidate Lengths

## Abstract

Speculative decoding reduces the inference latency of a target large language model via utilizing a smaller and faster draft model. Its performance depends on a hyperparameter $K$ — the candidate length, i.e., the number of candidate tokens for the target model to verify in each round. However, previous methods often use simple heuristics to choose $K$, which may result in sub-optimal performance. We study the choice of the candidate length $K$ and formulate it as a Markov Decision Process. We theoretically show that the optimal policy of this Markov decision process takes the form of a threshold policy, i.e., the current speculation should stop and be verified when the probability of getting a rejection exceeds a threshold value. Motivated by this theory, we propose `SpecDec++`, an enhanced version of speculative decoding that adaptively determines the candidate length on the fly. We augment the draft model with a trained acceptance prediction head to predict the conditional acceptance probability of the candidate tokens. `SpecDec++` will stop the current speculation when the predicted probability that *at least one token gets rejected* exceeds a threshold. We implement `SpecDec++` and apply it to the llama-2-chat 7B & 70B model pair. Our adaptive method achieves a 2.04x speedup on the Alpaca dataset (7.2% improvement over the baseline speculative decoding). On the GSM8K and HumanEval datasets, our method achieves a 2.26x speedup (9.4% improvement) and 2.23x speedup (11.1% improvement), respectively.

## 1 Introduction

Current state-of-the-art Large Language Models (LLMs) have demonstrated extraordinary capabilities in various language tasks and have shown early signs of artificial general intelligence (Achiam et al., 2023; Anil et al., 2023; Team et al., 2023; Touvron et al., 2023a;b). As the top-performing LLMs often have more than a hundred billion parameters, there is an increasing demand for serving such huge models efficiently. To decrease the inference latency, motivated by speculative execution techniques in processors, speculative decoding (Chen et al., 2023a; Leviathan et al., 2023) incorporates a **draft model**, which is smaller and faster, as the speculator for the **target model**, which is the large language model we want to accelerate. Given the current prefix, the draft model first auto-regressively generates $K$ tokens, taking substantially less time than it would take the target model. The target model computes their log probabilities *in parallel* and then sequentially determines whether each token is accepted or not. Following the first rejected token (if any), the algorithm discards the remaining tokens and corrects the rejected token with a fresh sample from a modified distribution. If all tokens are accepted, a new token is sampled from the next-token probability given by the target model and appended to the sequence of accepted tokens, and then the process moves forward. Such draft-verify-correct loops continue until the desired output is fully generated.

The speedup effect of speculative decoding depends on two crucial aspects: (1) how well the draft model aligns with the target model, and (2) how fast the draft model gets compared to the target model. The two aspects influence the choice of the hyperparameter $K$: the number of candidate tokens generated by the draft model in each loop. When the draft model aligns well and/or runs fast, we can choose a larger $K$, which potentially allows more tokens to be accepted in each loop. However, a larger $K$ also increases the chances of rejection so that more tokens get discarded.

Leviathan et al. (2023) studied the problem of choosing the hyperparameter $K$ under the assumption that the acceptance rates of all the candidate tokens are constant. The authors showed that there

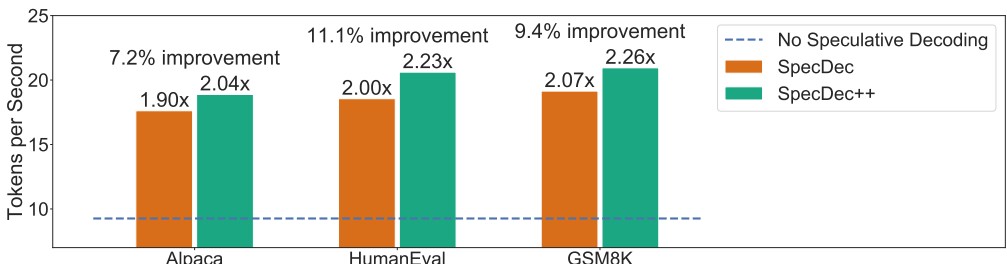

Figure 1: The performance of `SpecDec++`. Compared with the baseline speculative decoding (SpecDec) with fixed candidate lengths, by adaptively determining the candidate lengths via a trained acceptance prediction head, `SpecDec++` achieves a relative **7.2%**, **11.1%**, and **9.4%** improvement over the baseline methods on the Alpaca, HumanEval, and GSM8K dataset, respectively. The experiments are conducted with llama-2-chat 7B & 70B model pair on 2 NVIDIA A100-80G GPUs.

exists one constant $K$ that can maximize the speedup. However, such an assumption is unrealistic and does not approximate real-world cases well. Whether the draft model and the target model align well depends on the hardness of predicting the next token. Intuitively, when the next token is unambiguous from the prefix, the draft model and the target model align well, which means the acceptance probability of the current candidate token is large compared to other cases.

In this work, we aim to boost the performance of speculative decoding by adaptively choosing the candidate length $K$ for each round. We formalize the adaptive decision-making of $K$ for speculative decoding as a Markov Decision Process (MDP). The decision to make at each timestep is whether or not to stop the current speculation round and submit the candidate tokens to the target model for verification and correction. The objective is to minimize the total inference time taken to generate a full response. Theoretically, we show that the optimal policy takes the form of a threshold policy, i.e., it is optimal to stop the speculation round whenever the probability of existing at least one rejected token in the candidates exceeds a threshold.

Inspired by the theory, we propose `SpecDec++`, an enhanced version of speculative decoding that adaptively determines the candidate length on the fly. First, we train an acceptance prediction head on top of the draft model to predict the acceptance probability of the candidate token. Training such an acceptance prediction head has two challenges: (1) there will be a severe class imbalance problem, e.g., most tokens generated by the draft model will have a high probability of acceptance, depending on how well the two models align; (2) the input sequence to the model contains mostly tokens from the target model and only a fraction of tokens generated by the draft model, so the training signal is sparse. To overcome the two challenges, we adopt a weighted Binary Cross-Entropy loss to address the class imbalance problem, and we adapt the random masking idea from BERT (Devlin et al., 2019) to randomly mix tokens from the target model and the draft model to increase training efficiency.

At inference time, we opt to stop the current speculation round when the predicted probability of the existence of a rejected token exceeds a constant stopping threshold. The procedure is illustrated in Figure 2. We implement `SpecDec++` and apply it to llama-2-chat 7B & 70B model pair. Our adaptive method achieves a 2.04x speedup compared with the 1.90x speedup of the baseline speculative decoding method on the Alpaca dataset (an additional 7.2% improvement). On the easier GSM8K and HumanEval datasets, our method improves the baseline from 2.07x to 2.26x speedup (9.4% improvement) and from 2.00x to 2.23x speedup (11.1% improvement), respectively.

We summarize the contributions below.

- We formalize the dynamic choice of candidate length in speculative decoding as a Markov Decision Process (MDP). We theoretically show that when the probability that *at least one token gets rejected* exceeds a threshold, the optimal action is to stop the speculation and submit it for verification.
- We propose `SpecDec++`, an enhanced version of speculative decoding that adaptively determines the candidate length on the fly. We adopt a weighted loss and a token-mixing method to efficiently train the prediction head and use it for dynamic decision-making in the decoding process.
- Empirically, our method achieves an additional 7.2%, 9.4%, and 11.1% improvement over the baseline speculative decoding on the Alpaca, HumanEval, and GSM8K datasets, respectively.

## 2 BACKGROUND

**Rejection Sampling.** If we want to sample from a target discrete distribution $p(x)$, we first sample $x$ from a draft distribution $q(x)$. We accept the sample $x$ with probability $\min(1, \frac{p(x)}{q(x)})$; otherwise we replace it with a sample from the modified distribution $\text{Norm}[(p-q)_+]$, where $z_+ = \max(z, 0)$ is the positive part of $z$ and $\text{Norm}[f] = \frac{f(\cdot)}{\sum_x f(x)}$ normalizes a function $f$ to make it a proper probability distribution. The proof of the unbiasedness of rejection sampling can be found in (Chen et al., 2023a).

**Speculative Decoding.** Speculative decoding extends to the auto-regressive generation scenarios by chaining $K$ rejection sampling procedures together. To auto-regressively generate a sequence from $p(\cdot \mid x_{\text{prefix}})$, we first generates $K$ candidate tokens $(y_1, y_2, \ldots, y_K)$ from $q(\cdot \mid x_{\text{prefix}})$

$$y_i \sim q(Y_i \mid x_{\text{prefix}}, y_1, \ldots, y_{i-1}), \quad i = 1, 2, \ldots, K.$$

Next, we sequentially check if each $y_i$ is accepted or not. If there is any rejection, we replace the first rejected token with a fresh sample from the corresponding modified probability distribution and discard the subsequent tokens.

The key practical consideration is that the probabilities of the candidate tokens $p(y_i \mid x_{\text{prefix}}, y_1, \ldots, y_{i-1})$ can be calculated *in parallel* by the target model with no additional overhead, as the forward time is bottlenecked by the memory operations (Pope et al., 2023). For completeness, the speculative decoding algorithm is stated in Algorithm 1.

---

**Algorithm 1** Speculative Decoding (Chen et al., 2023a; Leviathan et al., 2023)

---

**Require:** draft model $q$, target model $p$, prefix $x_{\text{prefix}}$, number of candidate tokens $K$.
  **for** $i = 1$ to $K$ **do**
    Compute $q_i = q(\cdot \mid x_{\text{prefix}}, y_1, \ldots, y_{i-1})$.
    Sample $y_i \sim q_i$.
  **end for**
  Compute *in parallel* $p_i = p(\cdot \mid x_{\text{prefix}}, y_1, \ldots, y_{i-1})$ for $i = 1, \ldots, K+1$.
  Sample $r_1, \ldots, r_K$ with $r_i \sim \text{Unif}[0, 1]$, $i = 1, \ldots, K$.
  Compute the number of accepted tokens $n = \min \Big( \{i - 1 \mid r_i \geq p_i(y_i)/q_i(y_i)\} \cup K \Big)$.
  **if** $n < K$ **then**
    Sample $y'$ from the modified distribution $\text{Norm}[(p_{n+1} - q_{n+1})_+]$
  **else**
    Sample $y'$ from $p_{K+1}$
  **end if**
  **Return** $x_{\text{prefix}}, y_1, \ldots, y_n, y'$

---

**Inference Time of Speculative Decoding.**

Our objective is to minimize the total inference time, which consists of

$$T_{\text{total}} = t_{\text{draft}} N_{\text{draft}} + t_{\text{target}} N_{\text{target}}, \tag{2.1}$$

where $t_{\text{draft}}$ and $t_{\text{target}}$ are the time needed for one forward pass and $N_{\text{draft}}$ and $N_{\text{target}}$ are the total number of forward passes of the draft model and the target model, respectively. Equation (2.1) holds under the implicit assumption that the forward passes of each of the models take constant time, which is true when we have enough computational resources to support the increased concurrency when the length of the input sequence grows (Leviathan et al., 2023). We empirically verify that Equation (2.1) holds in our setting; see Section 4.2.

Let $N$ be the number of the final generated tokens. $N$ is a random variable inherent to the target model and the initial prompt, independent of the draft model and the number of candidate tokens $K$ of each round we choose. Let $N_{\text{discarded}}$ be the number of total discarded tokens. Then we have the following identity for Algorithm 1

$$N_{\text{draft}} + N_{\text{target}} = N + N_{\text{discarded}}.$$

Therefore, $T_{\text{total}}$ can be written as

$$T_{\text{total}} = T_0 + t_{\text{draft}} N_{\text{discarded}} + (t_{\text{target}} - t_{\text{draft}}) N_{\text{target}}, \tag{2.2}$$

where $T_0 = t_{\text{draft}} N$ is the oracle inference time.

To minimize the total inference time, we are required to trade-off between two objectives: minimizing the number of the discarded tokens $N_{\text{discarded}}$ and minimizing the number of forward passes of the target model $N_{\text{target}}$. The two objectives conflict with each other, as a larger $K$ will incur more discarded tokens but less number of forward passes of the target model. Equation (2.2) states that the total cost is the weighted sum of the two and the weights are given by $t_{\text{draft}}$ and $(t_{\text{target}} - t_{\text{draft}})$.

**Metrics.** To measure the benefit of a speculative decoding pipeline, we divide Equation (2.2) by $N$ and get

$$\text{latency} = T_{\text{total}}/N = t_{\text{draft}} + t_{\text{draft}} \cdot N_{\text{discarded}}/N + (t_{\text{target}} - t_{\text{draft}}) \cdot N_{\text{target}}/N. \tag{2.3}$$

We focus on two metrics: (1) **discard rate** $N_{\text{discarded}}/N$, which measures the average number of discarded tokens per one generated token, and (1) **verification rate** $N_{\text{target}}/N$, which measures the average number of the forward calls of the target model per one generated token.

### 2.1 A Motivating Example: Oracle Performances of Greedy Speculative Decoding

Let us focus on a simplified deterministic setting of speculative decoding, where we use greedy decoding for the draft model and the target model. In this setting, the draft model deterministically generates a series of greedy tokens $(Y_1, \ldots, Y_K)$, and the speculative decoding algorithm reduces to sequentially checking whether $Y_i$ is also the greedy token of the target model. The first rejected token is replaced by the greedy token of the target model. If all the tokens are accepted, an additional token is generated by the target model directly.

For a given prompt $x_{\text{prompt}}$, let $(X_1, X_2, \ldots, X_N)$ be the greedy tokens generated by the target model. We ask the following question:

*What is the oracle performance of the speculative decoding algorithm we can obtain by varying the number of candidate tokens, if we have the knowledge of $(X_1, X_2, \ldots, X_N)$ in hindsight?*

Let us consider the first speculation round. The draft model generates $(Y_1, Y_2, \ldots)$ greedily. Let $Y_i$ be the first token such that $Y_i \neq X_i$. The optimal strategy is to stop the speculation at time $(i-1)$, so the last candidate token $Y_{i-1}$ is accepted, and $Y_i$ will be generated directly by the target model, because (1) if we stop the speculation earlier, then the shorter candidate tokens will still be accepted, but this induces at least one unnecessary forward pass of the target model; (2) if we stop the speculation later, then we waste at least one candidate token $Y_i$. By repeating the argument, we have the following.

**Lemma 2.1.** In the greedy decoding setting, for a given prompt $x_{\text{prompt}}$, let $(X_1, X_2, \ldots, X_N)$ be the greedy tokens generated by the target model. We define $Y_i = \text{argmax}\, q(\cdot \mid x_{\text{prompt}}, X_1, X_2, \ldots, X_{i-1})$ to be the greedy token of the draft model $q$ conditioned on the partial generation of the target model. Let $S$ be the set of disagreement between the draft model and the target model: $S = \{1 \leq i \leq N \mid Y_i \neq X_i\}$. Then, by optimally stopping at time $(i-1)$ for every $i \in S$, we obtain the oracle performance with $N_{\text{discarded}} = 0$ and $N_{\text{target}} = |S| + 1$.

To empirically verify this, we perform a preliminary experiment with the same setting in Section 4, where we use all the prompts in the Alpaca dataset and calculate the set of disagreement $S$ for each prompt with the llama-2-chat-7B/llama-2-chat-70B model pair. The results show that the average $N_{\text{target}}/N = 0.164 \pm 0.078$ and the corresponding oracle throughput is $27.06 \pm 4.13$ tokens/second (2.92x speedup) according to Equation (2.3) with the empirical value of $(t_{\text{target}}, t_{\text{draft}})$ reported in Section 4.2. In comparison, the average throughput for the target model without speculative decoding is 9.26 tokens/second, while speculative decoding with the best fixed $K$ gives 17.58 tokens/second (1.90x speedup) (Section 4). We can see a huge potential in adaptively tuning the candidate lengths.

## 3 SpecDec++: Theory and Algorithm

### 3.1 Speculative Decoding as Markov Decision Processes

We formulate speculative decoding into the following Markov Decision Process (MDP) framework.

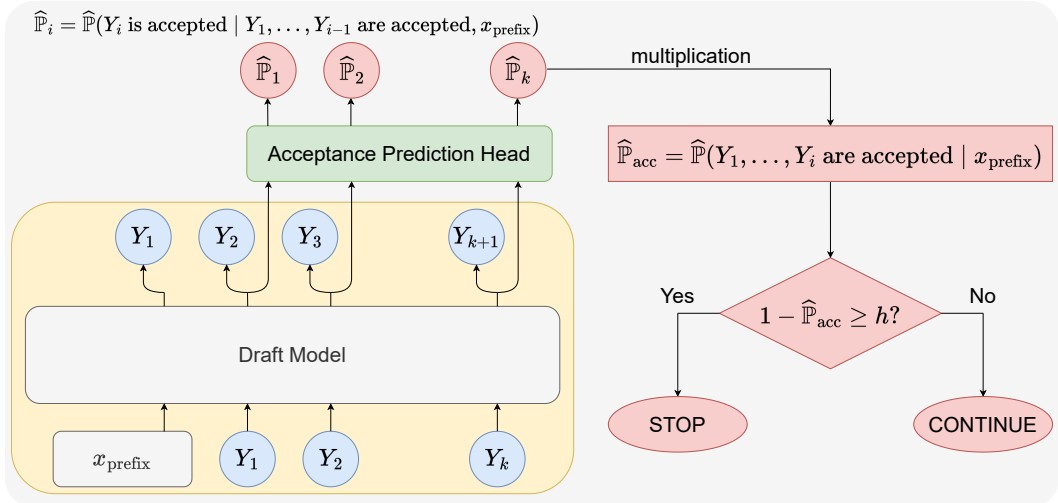

Figure 2: `SpecDec++` uses a trained **acceptance prediction head** to predict the conditional acceptance probability of the candidate tokens. When the predicted probability of the existence of at least one rejected token exceeds the **stopping threshold** $h$, the current speculation round ends and the candidate tokens go through the target model for verification and correction.

**States.** We define the tuple $s = (x_{\text{prefix}}, (Y_1, \ldots, Y_k))$ as the current state of the MDP. Specifically, $x_{\text{prefix}}$ is the concatenation of the prompt and the partial response containing all the accepted tokens. $(Y_1, \ldots, Y_k)$ is the current candidate tokens, which are auto-regressively sampled from the draft distribution $q$:

$$Y_i \sim q(\cdot \mid x_{\text{prefix}}, Y_1, \ldots, Y_{i-1}), \quad i = 1, 2, \ldots.$$

The initial state of the MDP is $(x_{\text{prompt}}, \varnothing)$.

**Actions.** Given the current state $(x_{\text{prefix}}, (Y_1, \ldots, Y_k))$, the decision to make is whether or not to end the current speculation round and submit the candidate tokens to the target model for verification. We denote the current action by $a \in \{\text{stop}, \text{continue}\}$ as the choice of stopping or continuing the current speculation round. [1]

We note that in an extended MDP setting, we can include the draft probability $q_{k+1}$ for the token $Y_{k+1}$ as a part of the current action. Finetuning the draft model to align better with the target model can be viewed as an offline policy optimization algorithm that will likely improve the performance. And it has been studied in previous work, e.g. DistillSpec (Zhou et al., 2024) and Medusa (Cai et al., 2024). In the paper, we consider the draft probability $q_{k+1}$ as given by the draft model and do not optimize $q_{k+1}$.

**Transitions.** First, we draw a random sample $Y_{k+1} \sim q_{k+1}$ and append $Y_{k+1}$ to the current list of the candidate tokens.

- When $a = \text{continue}$, the next state $s'$ is simply $(x_{\text{prefix}}, (Y_1, \ldots, Y_k, Y_{k+1}))$.
- When $a = \text{stop}$, the candidate tokens $(Y_1, \ldots, Y_{k+1})$ are verified via speculative decoding (Algorithm 1). Let $n$ be the number of the accepted tokens. Let $y'$ be the replaced token when $n < k + 1$ or the fresh token from the next-token distribution given by the target model when $n = k + 1$. The next state $s' = (x'_{\text{prefix}}, \varnothing)$ with the new prefix $x'_{\text{prefix}} = (x_{\text{prefix}}, y_1, \ldots, y_n, y')$ being the concatenation of the previous prefix and the newly generated tokens.

**Immediate Costs.** According to Equation (2.2), let $c_1 = t_{\text{draft}}$ and $c_2 = (t_{\text{target}} - t_{\text{draft}})$. We can define the immediate cost as the following

$$c(s, \text{continue}, s') = \mathbb{I}(\exists 1 \le i \le k + 1, Y_i \text{ is rejected}) \cdot c_1,$$

---

[1] In practice, when $Y_{k+1}$ is EOS (the special token denoting the end of sequence) or when the total length hits the maximal generation length, we manually set $a = \text{stop}$.

$$c(s, \text{stop}, s') = \mathbb{I}(\exists 1 \leq i \leq k+1, Y_i \text{ is rejected}) \cdot c_1 + c_2.$$

For both cases, we suffer a loss $c_1$ if the current candidate token $Y_{k+1}$ is discarded, which happens if there exists any candidate token $Y_i$ ($1 \leq i \leq k+1$) that is rejected. If we stop at the current step, we suffer an additional cost $c_2$ corresponding to the extra inference time of the target model.

Note that different from the traditional MDP setting when the reward/cost is immediately available to the learner, our setting is more related to the delayed feedback setting (Howson et al., 2023; Lee et al., 2023; Yang et al., 2024b; Chen et al., 2024a), where the cost is only available after the candidate tokens are submitted to the target model for verification.

**Theorem 3.1.** For any time-homogeneous policy $\pi$ that has an upper bound for the number of candidate tokens, at the current state $s = (x_{\text{prefix}}, (Y_1, \ldots, Y_k))$, when

$$\mathbb{P}(\exists 1 \leq i \leq k, Y_i \text{ is rejected} \mid x_{\text{prefix}}) \geq \frac{c_2 + \Delta}{c_1 + c_2 + \Delta},$$

the expected total cost of $\text{stop}$ is smaller than the expected total cost of $\text{continue}$, where $\Delta = \Delta(\pi, x_{\text{prompt}}, p, q, c_1, c_2)$ is a problem-specific constant.

We defer the proof of Theorem 3.1 to Appendix D.

### 3.2 SPECDEC++

Theorem 3.1 provides a sufficient condition for us to stop the current round of speculation and call the target model to verify the candidate tokens. Motivated by Theorem 3.1, we propose SpecDec++, an adaptive speculative decoding algorithm that utilizes an additional prediction head to determine whether or not to stop the current speculation round.

SpecDec++ incorporates an additional prediction head $f_\theta$ on top of the draft model that predicts the conditional probability

$$\mathbb{P}(Y_i \text{ is accepted} \mid Y_1, \ldots, Y_{i-1} \text{ are accepted}, x_{\text{prefix}}) = \min\left(1, \frac{p(Y_i | x_{\text{prefix}}, Y_1, \ldots, Y_{i-1})}{q(Y_i | x_{\text{prefix}}, Y_1, \ldots, Y_{i-1})}\right).$$

We opt to implement a small prediction head such that the computational overhead is negligible compared to a forward pass of the draft model. During inference time, we feed the input $(x_{\text{prefix}}, Y_1, \ldots, Y_i)$ to the draft model and obtain the final embedding $\boldsymbol{e}_i$ of the last token $Y_i$. The predicted acceptance probability is given by

$$\widehat{\mathbb{P}}(Y_i \text{ is accepted} \mid Y_1, \ldots, Y_{i-1} \text{ are accepted}, x_{\text{prefix}}) = \text{sigmoid}(f_\theta(\boldsymbol{e}_i)). \tag{3.1}$$

Given a threshold $h$, we end the current round of speculation when the predicted probability that there exists one rejected token exceeds $h$

$$\pi(s_k) = \text{stop} \Leftrightarrow \widehat{\mathbb{P}}(\exists 1 \leq i \leq k, \text{ such that } Y_i \text{ is rejected} \mid x_{\text{prefix}}) > h,$$

which can be computed by chain rule

$$\widehat{\mathbb{P}}(\exists 1 \leq i \leq k, \text{ such that } Y_i \text{ is rejected} \mid x_{\text{prefix}})$$

$$= 1 - \prod_{i=1}^{k} \widehat{\mathbb{P}}(Y_i \text{ is accepted} \mid Y_1, \ldots, Y_{i-1} \text{ are accepted}, x_{\text{prefix}}).$$

We summarize the proposed algorithm in Algorithm 2 and illustrate it in Figure 2.

### 3.3 TRAINING DATASET AND OBJECTIVE

Let $\mathcal{D}_{\text{prompt}}$ be the prompt distribution. For each $x_{\text{prompt}}$ in $\mathcal{D}_{\text{prompt}}$, we generate a response $(X_1, \ldots, X_N)$ using the target model. Next, we feed the prompt and the response into the draft model to get $q(\cdot \mid x_{\text{prompt}}, X_1, \ldots, X_{i-1})$ for every $i$. We sample a $Y_i$ from the distribution and calculate the conditional acceptance probability $\mathbb{P}_i = \min\left(1, \frac{p(Y_i | x_{\text{prompt}}, X_1, \ldots, X_{i-1})}{q(Y_i | x_{\text{prompt}}, X_1, \ldots, X_{i-1})}\right)$ for each token, which will be the training target.

**Algorithm 2** `SpecDec++`

---

**Require:** draft model $q$, target model $p$, prefix $x_{\text{prefix}}$, acceptance prediction head $f_\theta$, threshold $h$.

> **Initialize** the cumulative acceptance probability $\widehat{p} = 1$
> **for** $i = 1$ **do**
>    **if** $i > 1$ **then**
>      Compute the final hidden embedding $\boldsymbol{e}_{i-1}$ of the token $y_{i-1}$.
>    **end if**
>    Compute $q_i = q(\cdot \mid x_{\text{prefix}}, y_1, \ldots, y_{i-1})$.
>    Sample $y_i \sim q_i$.
>    Update $\widehat{p} \leftarrow \widehat{p} \cdot \text{sigmoid}(f_\theta(\boldsymbol{e}_{i-1}))$.
>    **if** $1 - \widehat{p} > h$ **then**
>      **Break**
>    **end if**
> **end for**

Let $K$ be the number of candidate tokens in the previous for-loop.
Compute *in parallel* $p_i = p(\cdot \mid x_{\text{prefix}}, y_1, \ldots, y_{i-1})$ for $i = 1, \ldots, K + 1$.
Sample $r_1, \ldots, r_K$ with $r_i \sim \text{Unif}[0, 1]$, $i = 1, \ldots, K$.

Compute the number of accepted tokens $n = \min \left( \{i - 1 \mid r_i \geq p_i(y_i)/q_i(y_i)\} \cup K \right)$.

**if** $n < K$ **then**
   Sample $y'$ from the modified distribution $\text{Norm}[(p_{n+1} - q_{n+1})_+]$
**else**
   Sample $y'$ from $p_{K+1}$
**end if**
**Return** $x_{\text{prefix}}, y_1, \ldots, y_n, y'$

---

We construct the response sequence $(Z_1, \ldots, Z_N)$ by randomly taking $r\%$ tokens from $(X_1, \ldots, X_N)$ and the remaining tokens from $(Y_1, \ldots, Y_N)$, borrowing the random masking idea from BERT (Devlin et al., 2019). We only compute losses for the tokens from $(Y_1, \ldots, Y_N)$.

We note that there will be distribution shifts between $(x_{\text{prefix}}, Y_1, \ldots, Y_k)$, the sequence encountered during the inference process, and $(x_{\text{prefix}}, Z_1, \ldots, Z_k)$, the sequence encountered during training process. The distribution shift may cause certain biases in the prediction head, e.g., over-confident about the acceptance. Furthermore, as in the typical setting of speculative decoding where the draft model and the target model align reasonably well, there will be class imbalance issues in the training dataset, where most of the training examples will have $\mathbb{P}_i$ close to 1.

To accommodate the issues above, we train the prediction head using a weighted binary cross-entropy (BCE) loss, taken over the tokens $Z_i$'s stemming from $Y_i$'s:

$$\sum_{x_{\text{prompt}} \in \mathcal{D}_{\text{prompt}}} \sum_{\substack{1 \leq i \leq N: \\ Z_i \text{ is taken from } Y_i}} \left( - w_{\text{acc}} \cdot \mathbb{P}_i \log \widehat{\mathbb{P}}_i - w_{\text{rej}} \cdot (1 - \mathbb{P}_i) \log(1 - \widehat{\mathbb{P}}_i) \right),$$

where $w_{\text{acc}}$ and $w_{\text{rej}}$ are the weights and $\widehat{\mathbb{P}}_i = \text{sigmoid}(f_\theta(\boldsymbol{e}_i(x_{\text{prompt}}, Z_1, \ldots, Z_{i-1}, Y_i)))$.

## 4 EXPERIMENTS

### 4.1 EXPERIMENTAL SETUPS

**Datasets and Model Pairs.** We adopt three datasets in our experiments: Alpaca (Taori et al., 2023), HumanEval (Chen et al., 2021), GSM8K (Cobbe et al., 2021). We only use prompts of the datasets and do not use responses. In the experiments, we use llama-2-chat models (Touvron et al., 2023b). We choose to use llama-2-chat 7B as the draft model and llama-2-chat 70B as the target model. To reduce memory consumption, we use the bfloat16 format for the models.

**Network Architecture, Weighted BCE Loss, and Stopping Criteria for `SpecDec++`.** We build a $(D + 1)$-layer ResNet with SiLU activation as the acceptance prediction head, and we sweep $D$

from 0 (linear layer) to 4 in the experiments. We adopt the weighted BCE loss where set $w_{\text{acc}} = 1$ and choose $w_{\text{rej}}$ from $\{1, 3, 6, 12\}$. We tune the stopping threshold $h$ in $\{0.1, 0.3, 0.5, 0.7, 0.9\}$. To ensure the robustness of `SpecDec++`, we manually stop each speculation round when the number of candidate tokens exceeds 20.

**Baseline Method.** We compare `SpecDec++` with the naive speculative decoding algorithm where the number of the candidate tokens $K$ is fixed as a hyperparameter. We tune $K$ in $\{2, 4, 6, 8, 10, 12, 14\}$.

Due to space limits, additional experimental setup is deferred to Appendix C.1.

## 4.2 FORWARD TIME ANALYSIS

First, we verify the correctness of Equation (2.1) and determine the forward time of the draft model $t_{\text{draft}}$ and the target model $t_{\text{target}}$ under our specific setting. We collect all the $(N_{\text{draft}}, N_{\text{target}}, T_{\text{total}})$ tuples from generations using speculative decoding (either the baseline version or `SpecDec++`) and perform a linear regression to determine the coefficients. We also determine the standalone inference time when using only the draft model or the target model with linear regression. The linear regressions fit well with all $R^2 \geq 0.98$ and the results are summarized in Table 2. Additionally, we visualize $t_{\text{draft}}$ and $t_{\text{target}}$ across the three settings in Figure 3.

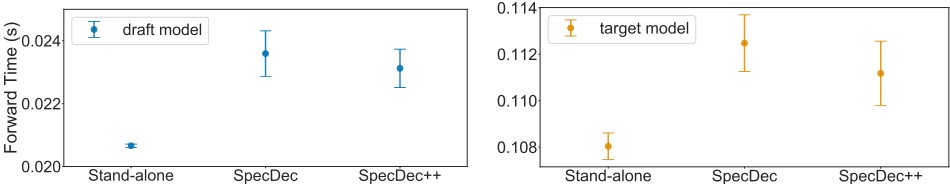

Figure 3: The forward time of the draft model (llama-2-chat-7B) and the target model (llama-2-chat-70B) under different settings. For each setting, we perform linear regression to calculate the forward times and then average them across different datasets. The additional cost of the acceptance prediction head is negligible compared to the systematic error and the random noise of the environment. Full results are deferred to Table 2.

**The additional cost of the acceptance prediction head is negligible**, as we find that the average $t_{\text{draft}}$ in `SpecDec++` setting is *smaller* than the average $t_{\text{draft}}$ in baseline SpecDec setting by $0.0004s$, which is likely caused by random noise of the environment, as the standard deviation between difference datasets around $0.0006s$. Therefore, for both the baseline speculative decoding setting and `SpecDec++` setting, we choose $(t_{\text{draft}}, t_{\text{target}}) = (0.0234, 0.112)$, which is the **average** between the two cases. We use Equation (2.3) to calculate the theoretical throughputs (tokens per second), which match the noisier empirical throughputs well with relative error $\leq 6.2\%$ for all prompts.

In the standalone setting where only the draft model or the target model is used, we see significant decreases in both $t_{\text{draft}}$ and $t_{\text{target}}$, which indicates that speculative decoding induces minor additional communication overhead. We use $(t_{\text{draft}}, t_{\text{target}}) = (0.0207, 0.108)$ for the stand-alone setting. The average throughput for the target model is 9.26 tokens/second.

## 4.3 PERFORMANCES

We test the performances of the baseline speculative decoding with different $K$ and `SpecDec++` with the different acceptance prediction heads and different thresholds $h$. We calculate the discard rates $N_{\text{discarded}}/N$ and the verification rates $N_{\text{target}}/N$ (Equation (2.3)). The results are plotted in Figure 4. We see that `SpecDec++` has strictly better Pareto frontiers than the baseline SpecDec on both the in-distribution test set Alpaca and the two out-of-distribution datasets HumanEval and GSM8K. Our method with adaptive candidate lengths improves upon the baseline method of fixed candidate lengths by reducing both the discard rate and the verification rate. The two metrics are **independent** of the actual forward times ($t_{\text{draft}}$ and $t_{\text{target}}$) and hence reusable for other hardware configurations, which indicates that `SpecDec++` will still outperform the baseline under different sets of $t_{\text{draft}}$ and $t_{\text{target}}$. Finally, we plug in the actual values of $(t_{\text{draft}}, t_{\text{target}}) = (0.0234, 0.112)$ as in Section 4.2. We summarize the throughputs in Table 1 and visualize the improvements in Figure 1.

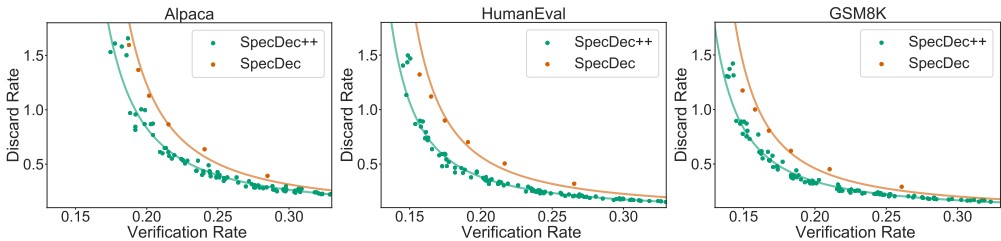

Figure 4: The average verification rates $N_{\text{target}}/N$ and the average discard rates $N_{\text{discarded}}/N$ for SpecDec with different candidate lengths and `SpecDec++` with different acceptance prediction heads and stopping thresholds. `SpecDec++` has better Pareto frontiers than SpecDec on both the in-distribution dataset Alpaca and the two out-of-distribution datasets HumanEval and GSM8K.

Table 1: The best throughputs achieved by `SpecDec++` compared to the best throughputs achieved by the speculative decoding baseline on Alpaca, HumanEval, and GSM8K datasets.

| Dataset | Alpaca | HumanEval | GSM8K |
|---|---|---|---|
| `SpecDec++` | 18.88 (tokens/s) | 20.61 (tokens/s) | 20.95 (tokens/s) |
| SpecDec (baseline) | 17.62 (tokens/s) | 18.55 (tokens/s) | 19.14 (tokens/s) |

**Discussions.** As the distribution shift of the OOD datasets will influence the accuracies and the calibrations of the acceptance prediction heads, a natural question to ask is whether the optimal performances for different datasets are achieved with different acceptance prediction heads and stopping thresholds. Empirically, we confirm that this is indeed the case. *Nevertheless*, we find that using the acceptance prediction trained with $w_{\text{rej}} = 6$ and network depth $D = 3$ and the stopping threshold $h = 0.7$ achieves over **99.3%** of the best tokens per second across the three datasets (2.03x for Alpaca, 2.21x for HumanEval, and 2.26x for GSM8K). Additional ablation studies on how the hyperparameters $(w_{\text{rej}}, D, h)$ influence the final tokens per second can be found in Appendix C.3.

## 5    RELATED WORK

**Speculative decoding.** Since the proposal of speculative decoding, people have been improving the algorithm from different perspectives. Our work is *complementary to* the works that improve speculative decoding by (1) making the draft model align better with the target model (Zhou et al., 2024; Agarwal et al., 2024; Liu et al., 2023), (2) building smaller draft models or merging draft models into the target model (e.g. early-exiting) (Miao et al., 2023; Liu et al., 2024; Yang et al., 2023b; Bae et al., 2023; Zhang et al., 2024; Monea et al., 2023; Chen et al., 2023b), and (3) building a heirachical system of speculative decoding (Spector & Re, 2023; Sun et al., 2024a). Our work is *not directly appliable to* the methods that do not have the concept of an auto-regressive draft model (Stern et al., 2018; Li et al., 2024b; Bhendawade et al., 2024; Cai et al., 2024) and the retrieval-based methods (He et al., 2023; Zhao et al., 2024; Yang et al., 2023a; Fu et al., 2024). See Appendix B for an extended related work about speculative decoding, token trees, and diffusion language models.

**Candidate length selection.** Leviathan et al. (2023) make the i.i.d. assumption on the acceptance probabilities of the candidate tokens and theoretically derive the optimal choice of $K$. Besides, Liu et al. (2024) and Kim et al. (2024) adopt a simple heuristic that ends the speculation if the confidence of the current draft token distribution falls below a threshold. Xu et al. (2023) uses the cumulative product of the confidences and extends to the token tree version. In comparison, our work systematically studies the candidate length selection within the MDP framework and uses the cumulative product of our trained prediction head to determine the end of the speculation.

## 6    CONCLUSION

We study the determination of the candidate lengths for speculative decoding. We formulate the problem as a Markov Decision Process and provide a theorem that gives a sufficient condition to stop the current speculation. Motivated by the theoretical result, we propose `SpecDec++` to adaptively select the candidate length with a trained acceptance prediction head. We demonstrate significant speedups over baselines and our method can be seamlessly integrated with other improvements.

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

## A  LIMITATIONS

Our theoretical result contains a problem-specific constant $\Delta$ which is hard to analyze theoretically or estimate empirically. Nevertheless, the choice of the stopping threshold $h$ can be determined through hyperparameter search; see Appendix C.3. As is the case with all speculative decoding algorithms, our method relies on the implicit assumption that the draft model and the target model align well. For a weak draft model, the acceptance prediction head may perform badly.

## B  ADDITIONAL RELATED WORK

Large language models are mostly based on Transformer architectures (Vaswani et al., 2017) that auto-regressively predict the probability of the next token given its predecessors. One bottleneck of the inference speed lies in the fact that auto-regressive decoding is an inherently non-parallelizable sequential operation: the probabilities of future tokens depend on the current token and there is no trivial way to skip the current token when predicting future tokens. Therefore, the inference time of auto-regressive decoding scales linearly with the number of the generated tokens.

However, the time of a forward pass to compute the log probabilities of the tokens through transformers is nearly constant for batched sequences with different lengths within a proper range, thanks to the increasingly powerful parallel computing units (Pope et al., 2023; Vaswani et al., 2017; Chen et al., 2023a; Leviathan et al., 2023).

Therefore, to overcome the bottleneck of the auto-regressive decoding, one can find a fast way to generate $K$ tokens, which often increases FLOPs, and the ask the target model to verify and correct the candidates (Stern et al., 2018; Chen et al., 2023a; Leviathan et al., 2023); see a comprehensive survey (Xia et al., 2024). For those methods to work, we assume that we have enough computational resources (e.g. CUDA memories) to support the increased concurrency. Nevertheless, in the long-context generation regime, the memory issue becomes prominent, which requires additional KV-cache management techniques such as compression or retrieval (Li et al., 2024a; Sun et al., 2024a).

**Improvements of Speculative Decoding Methods**

The performance of speculative decoding depends on how well the draft model aligns with the target model, and how fast the draft model is compared to the target model. People have been improving speculative decoding in two aspects: (1) making the draft model align better with the target model via distillation (Zhou et al., 2024; Agarwal et al., 2024) and online learning (Liu et al., 2023); and (2) making the token generation faster and cheaper, e.g. training multiple smaller draft models from stratch (Miao et al., 2023).

In addition, the candidate tokens can be generated without a separate draft model (Stern et al., 2018; Li et al., 2024b; Du et al., 2024; Bhendawade et al., 2024), such as building additional modules that predict the next $k$ tokens (Medusa heads (Cai et al., 2024), RNN heads (Zhang et al., 2024), soft tokens (Monea et al., 2023)), early-exiting methods that reuse the intermediate representations of the target model (Liu et al., 2024; Yang et al., 2023b; Bae et al., 2023), and retrieval-based methods that involve constructing an $n$-gram datastore and using retrieval to generate candidates (He et al., 2023; Zhao et al., 2024; Yang et al., 2023a; Fu et al., 2024).

Those techniques can be combined, resulting in a heirachical system (Spector & Re, 2023; Zhao et al., 2024; Sun et al., 2024a).

**Token Tree Generation, Verification and Pruning.**

Paralleling across the batch dimension via token trees is another direction to increase throughputs (Miao et al., 2023; Xu et al., 2023; Su et al., 2023). For greedy decoding, token tree generation and verification are studied in (Cai et al., 2024). For the stochastic sampling setting, REST (He et al., 2023) proposes a straightforward approach: keeping the token paths that coincide with the stochastic tokens given by the target model. There are also researches extending the stochastic speculative decoding to the token tree setting, which often needs to adjust the drafting and verification probabilities to ensure unbiasedness, e.g. MCSD (Yang et al., 2024a), Recursive SD (Jeon et al., 2024), Sequoia (Chen et al., 2024b), EAGLE (Li et al., 2024b), SpecTR (Sun et al., 2024b).

One important problem to study is how to construct and prune the token tree to maximize throughputs and avoid heavy communication overheads, which is studied in (Chen et al., 2024b; Zhong et al., 2024). Our work can serve as a starting point towards the problem, as the candidate length $K$ can be viewed as the depth of a token tree with only one branch.

**Diffusion language models.** Diffusion language models either in the discrete space (see D3PM (Austin et al., 2021) and its follow-ups) or in the embedding space (see Diffusion-LM (Li et al., 2022) and its follow-ups) are non-autoregressive language models, whose generation time can scale sub-linearly with the sequence length. BERT-type encoder-only models and auto-regressive decoder-only models can be also viewed as diffusion model, with mask prediction and next-token prediction being the denoising operation (Austin et al., 2021). Viewing next-token prediction as *Jacobi iteration* (Santilli et al., 2023) and *denoising operation* is a powerful idea and it leads to subsequent work such as lookahead decoding (Fu et al., 2024) and consistency LLMs (Kou et al., 2024).

## C ADDITIONAL EXPERIMENTAL RESULTS

### C.1 ADDITIONAL EXPERIMENTAL SETUPS

The subsection continues Section 4.1.

**Datasets.** We adopt three datasets in our experiments: (1) Alpaca (Taori et al., 2023), an instruction-following dataset generated using Self-Instruct (Wang et al., 2023) from OpenAI's `text-davinci-003` model; (2) HumanEval (Chen et al., 2021), a test dataset containing Python code synthesis problems; and (3) GSM8K (Cobbe et al., 2021), a dataset of high-school math problems. We only use prompts of the datasets and do not use responses.

**Dataset splits.** We split the Alpaca dataset into train/dev/test splits, containing 40k, 10k, 2k prompts, respectively. We use train split to train the prediction heads and evaluate them on the dev split. We benchmark the performance of `SpecDec++` on the test split. For HumanEval and GSM8K, we only use them for benchmarking the out-of-distribution (OOD) performance of `SpecDec++`. For each test dataset, we subsample 150 examples for benchmarking the performances.

**Mixing probability.** As in Section 3.3, we mix the response tokens from the generations from the target model and the predicted next-tokens from the draft model. We set an aggressive value $r\% = 15\%$ so only 15% of the tokens are from the target model, as we find empirically that the draft model and the target model often align well. Setting a smaller $r$ increases the training efficiency as more supervision signals are used.

**Training Details.** We train all the acceptance prediction heads on the train split of the Alpaca dataset for 3 epochs with batch size 32. We use Adam optimizer and a cosine learning rate schedule with the initial learning rate $5e - 5$.

**Hardware configuration.** We use 2 NVIDIA A100 GPUs with 80G memory for the experiments. We shard the 70B model across the two devices and communication overhead occurs when inferring with llama-2-chat 70B. When doing speculative decoding, the 7B model is loaded only on one device.

**Inference setting.** We set the maximal sequence length to be 512. We use temperature $T = 1$ and adopt top-k sampling with $k = 50$. We do not integrate KV cache management techniques such as PagedAttention (Kwon et al., 2023) or KV cache pre-allocation.

**Experiments Compute Resources.** The required compute resources are estimated to be 500 hours on 2 NVIDIA A100-80G GPUs for the training dataset generation, 400 hours on 1 NVIDIA A100-80G GPU for training 20 acceptance prediction heads (sweeping $D$ from 0 to 4 and $w_{\text{rej}}$ among $1, 3, 6, 12$), 500 hours on 2 NVIDIA A100-80G GPUs for the whole evaluation set. The full research project would require at least 2x the reported compute, as there were preliminary experiments that are not in the paper.

### C.2 FORWARD TIME ANALYSIS

We report the full results of the linear regression in Section 4.2 in Table 2.

Table 2: The forward time of the draft model (llama-2-chat-7B) and the target model (llama-2-chat-70B) under different settings and different datasets. We perform linear regression to calculate the forward times.

| Setting | Dataset | $t_{\text{draft}}$ | $t_{\text{target}}$ | $R^2$ |
|---|---|---|---|---|
| stand-alone | Alpaca | 0.0206 | 0.108 | 0.9994 & 0.9998 |
| | HumanEval | 0.0207 | 0.107 | 0.9994 & 0.9998 |
| | GSM8K | 0.0206 | 0.109 | 0.9990 & 0.9992 |
| | average | $0.0207 \pm 0.0001$ | $0.108 \pm 0.001$ | |
| SpecDec | Alpaca | 0.0232 | 0.114 | 0.9983 |
| | HumanEval | 0.0246 | 0.111 | 0.9965 |
| | GSM8K | 0.0229 | 0.113 | 0.9926 |
| | average | $0.0236 \pm 0.0007$ | $0.112 \pm 0.001$ | |
| SpecDec++ | Alpaca | 0.0240 | 0.110 | 0.9982 |
| | HumanEval | 0.0229 | 0.111 | 0.9880 |
| | GSM8K | 0.0225 | 0.113 | 0.9925 |
| | average | $0.0231 \pm 0.0006$ | $0.111 \pm 0.001$ | |

### C.3 ABLATION STUDIES.

We study how the hyperparameters $w_{\text{rej}}, D, h$ influence the final throughputs (tokens per second). First, we calculate the (unweighted) binary KL divergence between the ground-truth probability and the predicted probability, i.e.,

$$\text{KL}(p||q) = p \log \frac{p}{q} + (1-p) \log \frac{1-p}{1-q}.$$

As $\text{KL}(p||q) = \text{BCE}(p||q) - H(p)$, the binary KL divergence is a metric for how well the acceptance prediction head fits the ground-truth probabilities. Next, for each acceptance prediction head, we report the best throughput by varying the stopping threshold $h$ among $\{0.1, 0.3, 0.5, 0.7, 0.9\}$, and the corresponding $h$ that achieves the best performance. The results are summarized in Table 3.

From the table, we see that increasing $w_{\text{rej}} = 1$ increases the *unweighted* eval KL. All the prediction heads trained with $w_{\text{rej}} = 1$ perform the best with $h = 0.3$ under all three datasets, and similarly, most prediction heads trained with $w_{\text{rej}} = 3, 6, 12$ perform the best with $h = 0.5, 0.7, 0.9$, respectively. This synergy between $w_{\text{rej}} = 1$ and $h$ is expected, since increasing $w_{\text{rej}} = 1$ forces the acceptance prediction head to focus more on the cases where the candidate token is rejected and thus mitigates the over-confidence issue. In return, the stopping threshold $h$ can be set to a higher value to adjust for the increased predicted probability of existing one rejection.

We bold the throughputs that are above 99% of the maximum throughput of the same dataset. We see that there are two sets of hyperparameters that consistently achieve 99% of the maximum throughputs across the three datasets: $w_{\text{rej}} = 6, D = 3, h = 0.7$ and $w_{\text{rej}} = 6, D = 4, h = 0.7$.

## D THEORETICAL ANALYSIS

In the section, we present the proof of Theorem 3.1.

For any time-homogeneous policy $\pi$, we define a random variable $C^\pi(s, a)$ as the total cost-to-go from the current state $s = (x_{\text{prefix}}, (Y_1, \ldots, Y_k))$ when taking action $a$.

$$C^\pi(s, a) = \sum_{i=1}^{M} c(s_i, a_i, s_{i+1}), \text{ with } s_1 = s, a_1 = a,$$

where the next state $s_{i+1}$ given $(s_i, a_i)$ follows the stochastic transition of the MDP, $a_i = \pi(s_i)$ for $i \geq 2$, and $M$ is a random variable of the number of total steps. We make the assumption that $\pi$ has an upper bound for the number of candidate tokens, so we exclude the cases where the policy $\pi$ potentially leads to an infinite loop and hence $M < \infty$. Let $C^\pi(s) = C^\pi(s, \pi(s))$.

*proof of Theorem 3.1.* We analyze the difference $C^\pi(s, \text{continue}) - C^\pi(s, \text{stop})$ for three cases.

Table 3: The performance of the acceptance prediction heads with different loss weights $w_{\text{rej}}$ and network depths $D$. The train/eval KL refers to the binary KL divergence between the ground-truth probability and the predicted probability. For the three datasets, we report the best throughput and the corresponding stopping threshold $h$. The throughputs are **bolded** if they are above 99% of the maximum throughput of the same dataset.

| $w_{\text{rej}}$ | Depth $D$ | train/KL | eval/KL | Alpaca | HumanEval | GSM8K |
|---|---|---|---|---|---|---|
| 1 | 0 | 0.422 | 0.412 | 18.48 ($h = 0.3$) | 19.91 ($h = 0.5$) | 20.32 ($h = 0.3$) |
| 1 | 1 | 0.409 | 0.390 | 18.39 ($h = 0.3$) | 20.29 ($h = 0.3$) | 20.44 ($h = 0.3$) |
| 1 | 2 | 0.391 | 0.387 | **18.87** ($h = 0.3$) | 20.26 ($h = 0.3$) | **20.87** ($h = 0.3$) |
| 1 | 3 | 0.387 | 0.384 | **18.82** ($h = 0.3$) | 20.10 ($h = 0.3$) | **20.86** ($h = 0.3$) |
| 1 | 4 | 0.384 | 0.383 | 18.57 ($h = 0.3$) | **20.51** ($h = 0.3$) | 20.73 ($h = 0.3$) |
| 3 | 0 | 0.515 | 0.491 | 18.31 ($h = 0.5$) | 20.12 ($h = 0.7$) | 20.36 ($h = 0.5$) |
| 3 | 1 | 0.479 | 0.461 | **18.88** ($h = 0.5$) | 20.32 ($h = 0.5$) | 20.70 ($h = 0.5$) |
| 3 | 2 | 0.475 | 0.458 | 18.60 ($h = 0.5$) | 20.17 ($h = 0.5$) | 20.61 ($h = 0.3$) |
| 3 | 3 | 0.462 | 0.454 | **18.76** ($h = 0.5$) | 20.32 ($h = 0.5$) | **20.88** ($h = 0.5$) |
| 3 | 4 | 0.465 | 0.451 | **18.88** ($h = 0.5$) | **20.50** ($h = 0.7$) | **20.82** ($h = 0.5$) |
| 6 | 0 | 0.657 | 0.637 | 18.67 ($h = 0.7$) | 19.90 ($h = 0.9$) | 20.24 ($h = 0.7$) |
| 6 | 1 | 0.620 | 0.596 | **18.75** ($h = 0.7$) | 20.09 ($h = 0.9$) | **20.86** ($h = 0.7$) |
| 6 | 2 | 0.607 | 0.589 | 18.65 ($h = 0.7$) | 20.17 ($h = 0.9$) | 20.70 ($h = 0.7$) |
| 6 | 3 | 0.617 | 0.582 | **18.80** ($h = 0.7$) | **20.47** ($h = 0.7$) | **20.95** ($h = 0.7$) |
| 6 | 4 | 0.603 | 0.575 | **18.87** ($h = 0.7$) | **20.61** ($h = 0.7$) | **20.77** ($h = 0.7$) |
| 12 | 0 | 0.922 | 0.871 | 18.55 ($h = 0.9$) | 19.93 ($h = 0.9$) | 20.62 ($h = 0.9$) |
| 12 | 1 | 0.830 | 0.805 | **18.71** ($h = 0.9$) | 20.25 ($h = 0.9$) | 20.73 ($h = 0.9$) |
| 12 | 2 | 0.834 | 0.794 | 18.58 ($h = 0.9$) | 20.39 ($h = 0.9$) | **20.77** ($h = 0.7$) |
| 12 | 3 | 0.801 | 0.781 | **18.76** ($h = 0.9$) | 20.29 ($h = 0.9$) | 20.67 ($h = 0.9$) |
| 12 | 4 | 0.799 | 0.773 | **18.82** ($h = 0.9$) | 20.19 ($h = 0.9$) | 20.65 ($h = 0.9$) |

**Case 1.** $\mathcal{E}_1 = \{\exists 1 \leq i \leq k + 1, \text{ such that } Y_i \text{ is rejected}\}$.

Let $x'_{\text{prefix}}$ be the next prefix given by the speculative decoding algorithm, where the first rejected token among $(Y_1, \ldots, Y_{k+1})$ is replaced by the token from the modified distribution. We know that

$$C^{\pi}(s, \text{stop}) = c_1 + c_2 + C^{\pi}((x'_{\text{prefix}}, \varnothing)).$$

If we choose to continue at the current step, we know that no matter how many additional steps we continue to generate draft tokens, we will eventually discard them and get the same new prefix $x'_{\text{prefix}}$. Let $N^{\pi}_{\text{continue}}(s)$ be the total number of extra continue's induced by the policy $\pi$ given the current state $s$ and action continue. We have

$$C^{\pi}(s, \text{continue}) = c_1 + c_1 \cdot (1 + N^{\pi}_{\text{continue}}(s)) + c_2 + C^{\pi}((x'_{\text{prefix}}, \varnothing)).$$

In summary, we have

$$C^{\pi}(s, \text{continue}) - C^{\pi}(s, \text{stop}) \geq c_1, \text{ conditioned on } \mathcal{E}_1.$$

**Case 2.** $\mathcal{E}_2 = \{\forall 1 \leq i \leq k + 1, Y_i \text{ is accepted}, Y_{k+2} \text{ is rejected}\}$.

If we stop the current round of speculation, then all the candidate tokens $(Y_1, \ldots, Y_{k+1})$ will be accepted and an additional $X_{k+2}$ is sampled from $p(\cdot \mid x_{\text{prefix}}, Y_1, \ldots, Y_{k+1})$.

$$C^{\pi}(s, \text{stop}) = c_2 + C^{\pi}(((x_{\text{prefix}}, Y_1, \ldots, Y_{k+1}, X_{k+2}), \varnothing)).$$

Again, if we choose to continue at the current step, as $Y_{k+2}$ is rejected, future generated tokens beyond $Y_{k+2}$ will also be discarded. After the verification, $Y_{k+2}$ will be replaced by $W_{k+2} \sim \text{Norm}[(p(\cdot|x_{\text{prefix}}, Y_1 \ldots, Y_{k+1}) - q(\cdot|x_{\text{prefix}}, Y_1 \ldots, Y_{k+1}))_+]$. Let $N^{\pi}_{\text{continue}}(s)$ be the total number of extra continue's induced by the policy $\pi$ given the current state $s$ and action continue. We have

$$C^{\pi}(s, \text{continue}) = c_1 \cdot (1 + N^{\pi}_{\text{continue}}(s)) + c_2 + C^{\pi}(((x_{\text{prefix}}, Y_1, \ldots, Y_{k+1}, W_{k+2}), \varnothing)).$$

Denote $\Delta_1 = C^{\pi}(((x_{\text{prefix}}, Y_1, \ldots, Y_{k+1}, X_{k+2}), \varnothing)) - C^{\pi}(((x_{\text{prefix}}, Y_1, \ldots, Y_{k+1}, W_{k+2}), \varnothing))$. In summary, we have

$$C^{\pi}(s, \text{continue}) - C^{\pi}(s, \text{stop}) \geq c_1 - \Delta_1, \text{ conditioned on } \mathcal{E}_2.$$

**Case 3.** $\mathcal{E}_3 = \{\forall 1 \leq i \leq k+2, Y_i \text{ is accepted}\}$.

Similar to Case 2, if we stop the current round of speculation, then all the candidate tokens $(Y_1, \ldots, Y_{k+1})$ will be accepted, and an additional $X_{k+2}$ is sampled from $p(\cdot \mid x_{\text{prefix}}, Y_1, \ldots, Y_{k+1})$.

$$C^\pi(s, \text{stop}) = c_2 + C^\pi(((x_{\text{prefix}}, Y_1, \ldots, Y_{k+1}, X_{k+2}), \varnothing)).$$

If we choose to continue at the current step, there is no immediate cost at the current step and we transit to $(x_{\text{prefix}}, (Y_1, \ldots, Y_{k+1}))$.

$$C^\pi(s, \text{continue}) = C^\pi((x_{\text{prefix}}, (Y_1, \ldots, Y_{k+1}))).$$

Denote $\Delta_2 = C^\pi(((x_{\text{prefix}}, Y_1, \ldots, Y_{k+1}, X_{k+2}), \varnothing)) - C^\pi((x_{\text{prefix}}, (Y_1, \ldots, Y_{k+1})))$. We have

$$C^\pi(s, \text{continue}) - C^\pi(s, \text{stop}) \geq -c_2 - \Delta_2, \text{ conditioned on } \mathcal{E}_3.$$

**Summary.** At the current state, the values of $(Y_1, \ldots, Y_k)$ are known. We calculate the conditional expectation of $C^\pi(s, \text{continue}) - C^\pi(s, \text{stop})$ given the current observation. For simplicity of notation, we do not explicitly write out the condition on $(Y_1, \ldots, Y_k)$.

$$\mathbb{E}[C^\pi(s, \text{continue}) - C^\pi(s, \text{stop})]$$
$$\geq \mathbb{P}(\mathcal{E}_1)c_1 + \mathbb{P}(\mathcal{E}_2)(c_1 - \mathbb{E}[\Delta_1 \mid \mathcal{E}_2]) + \mathbb{P}(\mathcal{E}_3)(-c_2 - \mathbb{E}[\Delta_2 \mid \mathcal{E}_3]).$$

When the right-hand side of the above inequality is larger than zero, the expected total cost of continue is larger than the expected cost of stop. Therefore, we obtain a sufficient condition to stop at the current step.

To continue the analysis, we assume that we have an almost-sure upper bound $\Delta$ on $\mathbb{E}[\Delta_1 \mid \mathcal{E}_2]$ and $\mathbb{E}[\Delta_2 \mid \mathcal{E}_3]$:

$$\mathbb{E}[\Delta_1 \mid \mathcal{E}_2] \leq \Delta \ a.s. \text{ and } \mathbb{E}[\Delta_2 \mid \mathcal{E}_3] \leq \Delta \ a.s..$$

A naive bound for $\Delta$ is the upper bound of $C$, e.g., $\max N_{\text{target}} \cdot t_{\text{target}} + \max N_{\text{draft}} \cdot t_{\text{draft}}$. We assume that both the maximum generated tokens and the numbers of candidate tokens per round have an upper limit, so the upper bound is finite.

Then

$$\mathbb{P}(\mathcal{E}_1)c_1 + \mathbb{P}(\mathcal{E}_2)(c_1 - \mathbb{E}[\Delta_1 \mid \mathcal{E}_2]) + \mathbb{P}(\mathcal{E}_3)(-c_2 - \mathbb{E}[\Delta_2 \mid \mathcal{E}_3]) \geq 0$$
$$\Leftrightarrow \quad \mathbb{P}(\mathcal{E}_1)c_1 + \mathbb{P}(\mathcal{E}_2)c_1 \geq \mathbb{P}(\mathcal{E}_3)c_2 + \mathbb{P}(\mathcal{E}_3)\mathbb{E}[\Delta_2 \mid \mathcal{E}_3] + \mathbb{P}(\mathcal{E}_2)\mathbb{E}[\Delta_1 \mid \mathcal{E}_2]$$
$$\Leftarrow \quad \mathbb{P}(\mathcal{E}_1)c_1 + \mathbb{P}(\mathcal{E}_2)c_1 \geq \mathbb{P}(\mathcal{E}_3)c_2 + \mathbb{P}(\mathcal{E}_3)\Delta + \mathbb{P}(\mathcal{E}_2)\Delta$$
$$\Leftarrow \quad \mathbb{P}(\mathcal{E}_1)c_1 \geq (\mathbb{P}(\mathcal{E}_2) + \mathbb{P}(\mathcal{E}_3))c_2 + (\mathbb{P}(\mathcal{E}_3) + \mathbb{P}(\mathcal{E}_2))\Delta$$
$$\Leftrightarrow \quad \mathbb{P}(\mathcal{E}_1) \geq \frac{c_2 + \Delta}{c_1 + c_2 + \Delta}.$$

Finally, we note that

$$\mathbb{P}(\mathcal{E}_1) = \mathbb{P}[\exists 1 \leq i \leq k+1, \text{ such that } Y_i \text{ is rejected} \mid Y_1, \ldots, Y_k]$$
$$\geq \mathbb{P}[\exists 1 \leq i \leq k, \text{ such that } Y_i \text{ is rejected} \mid Y_1, \ldots, Y_k],$$

which concludes the proof. $\qquad\square$

