# OpenReview forum: "SpecDec++: Boosting Speculative Decoding via Adaptive Candidate Lengths"
_ICLR.cc/2025/Conference — Submitted to ICLR 2025_

### Official Review · Reviewer_EdTr · 2024-10-28

**Soundness:** 3
**Presentation:** 3
**Contribution:** 2
**Rating:** 5
**Confidence:** 4

**Summary:**

This paper introduces a new speculative decoding method called SpecDec++, whose main contributions are formalizing the candidate length selection problem as a Markov Decision Process (MDP) and dynamically adjusting the number of candidate tokens. This is achieved through a well-trained acceptance prediction head that predicts the conditional probability of candidate tokens being accepted by the target model. The method demonstrates substantial speedup over baseline speculative decoding approaches with fixed candidate lengths.

**Strengths:**

1、Speculative decoding is of significant importance for improving the inference efficiency of large language models.

2、The paper provides sufficient justification in theoretical analysis by modeling the problem as an MDP and proposing a threshold policy, which offers a solid theoretical foundation for dynamically adjusting the candidate length.

3、By reducing the number of forward passes of the target model and decreasing the number of discarded tokens, SpecDec++ achieves faster inference speed compared to the baseline.

**Weaknesses:**

1、The paper aims to accelerate inference by reducing the number of forward passes of the target model and decreasing the number of discarded tokens, but the experiments show only a modest improvement over the baseline. Moreover, the training cost for the head is substantial. I wonder if these two aspects are not the key bottlenecks of speculative sampling, or if this method is not very effective in addressing these issues. This makes me somewhat skeptical about the effectiveness of this approach.

2、The paper introduces two evaluation metrics to demonstrate the effect of dynamically adjusting the candidate token length, but the length of accepted tokens is also a crucial metric. I am not sure if this method has any impact on the length of accepted tokens.

3、The paper only presents results on the llama-2-chat 7B and llama-2-chat 70B models. I suspect this is due to the high training costs, but a single set of results is less convincing.

**Questions:**

1、Are there other methods to reduce the training cost, or are there alternative approaches to determine whether to stop the draft model's generation?

2、The paper claims that the method can be seamlessly integrated with other improvements. Could at least one example be provided for experimentation?

3、It would be desirable to conduct experiments on a broader range of models.

---

> ### Author Response · Authors · 2024-11-21
>
> We sincerely appreciate the time and effort you have invested in reviewing our paper and providing valuable feedback. We would like to provide detailed responses below:
>
> ----
>
> > **W1**:The paper aims to accelerate inference by reducing the number of forward passes of the target model and decreasing the number of discarded tokens, but the experiments show only a modest improvement over the baseline. Moreover, the training cost for the head is substantial.  .... This makes me somewhat skeptical about the effectiveness of this approach.
>
> **A1**: We would like to re-emphasize the practical contribution of our method:
>
> - (1) The speedup is achieved through **algorithmic improvements** as opposed to system-level improvements. This means that our method can be plugged in whenever the inference method adopts a form of draft models (e.g., early-exiting).  Furthermore, it can be readily combined with other system-level and hardware-level improvements, e.g., fused cuda kernels, smarter KV cache management, or simply faster chips.
> - (2) Our experimental results are **hardware-independent**: Figure 4 shows that our proposed method has better Pareto front than the baseline SpecDec method, which means SpecDec++ will still out-perform the baseline SpecDec under other hardware configurations.   One may argue that the key bottlenecks of speculative sampling is the system-level optimization tricks. Nevertheless, when those bottlenecks are solved, SpecDec++ will still achieve improvement over baseline SpecDec.
> - (3) Besides, achieving an additional 7-11% speed improvement will save 7%-11% of the GPU time and the **financial cost** for those entities that need to serve their LLMs, which could be a huge number for LLM applications with high traffic.
>
> **About training cost:** The training cost for the prediction head is **marginal** as we only tune a scalar output head **on top of** the last layer of hidden features of the draft model.
> - (1) As a comparison, Medusa [1] involves tuning multiple token prediction heads and the total network size of Medusa heads is larger than ours.
> - (2) In Appendix C.1, we have a rough estimate of the compute resources used in the paper. Training the acceptance head only takes **16.7%** of the total compute. In the real-world setting where we host the model for inference service, the one-time training cost will be essentially negligible compared to the vast amount of the **inference cost**.
>
> ----
>
> > **W2**: The paper introduces two evaluation metrics to demonstrate the effect of dynamically adjusting the candidate token length, but the length of accepted tokens is also a crucial metric. I am not sure if this method has any impact on the length of accepted tokens.
>
> **A2**: Thank you for indicating that “the length of accepted tokens” is also a crucial metric. Indeed, our method has a substantial impact on the average length of the consecutive accepted tokens. Actually, the “verification rate” in our paper is defined as the average number of the target model forward pass *per* accepted token. You can see that **1/verification rate** is the average length of the accepted tokens per round. By adaptively determining the draft token length, our method achieves a larger length of accepted tokens while keeping the number of discarded tokens controlled.
>
> ----
>
> > **W3**: The paper only presents results on the llama-2-chat 7B and llama-2-chat 70B models. I suspect this is due to the high training costs, but a single set of results is less convincing.
>
> **A3**: Thank you for pointing the issue out. We repeat all our experiments on a new pair of models: Gemma-1.1-2B-it and Gemma-2-27B-it. We are also able to show performance gains of our proposed SpecDec++ compared to the baseline SpecDec. Please see the [general response](https://openreview.net/forum?id=NnExMNiTHw&noteId=xIyrvcM1Nj) for details!
>
> Besides, the training cost is not high; please see the complete discussion in **A1**.
>
> ----
>
> ### Questions:
>
> > **Q1**: Are there other methods to reduce the training cost, or are there alternative approaches to determine whether to stop the draft model's generation?
>
> **A1**: The training cost is not high; please see the complete discussion in answer **A1** to **W1**.
>
> ---
>
> > **Q2**: The paper claims that the method can be seamlessly integrated with other improvements. Could at least one example be provided for experimentation?
>
> **A2**: Our experimental results utilize the half-precision (bf16) models to mitigate the memory I/O overhead for both the draft model and the target model, which is a widely adopted technique. Our method can be easily integrated with a more quantized draft model.
>
> ---
> > **Q3**: It would be desirable to conduct experiments on a broader range of models.
>
> **A3**: Please see the general response for the new experiments.
>
> ---
>
> **We sincerely hope that our responses can address your concerns, and we would greatly appreciate it if you would like to re-evaluate our work given the responses.**

---

> > ### Comment · Reviewer_EdTr · 2024-11-22
> >
> > The author's responses solved most of my questions. I saw that the author mentioned that it can be combined with EAGLE. I think it would be better if it could be supplemented with experiments on EAGLE or other methods, not limited to SpecDec.I will re-evaluate this work.

---

> > > ### Author Response · Authors · 2024-11-22
> > > **Thank you for your response**
> > >
> > > We are delighted that our reply solved most of your concerns. Indeed, combining our adaptive candidate length method with a stronger baseline will strengthen the presentation of the paper. However, given the limited time left, it is unfortunately hard for us to conduct a full study within the author-reviewer discussion time window. We will aim for the expanded study in our next revision.
> > >
> > > Still, we believe the contribution of the paper is the **relative improvements** over the baseline algorithm. When switching to a stronger baseline (e.g., early-exiting as a draft model, distilled draft model, cascading speculative decoding, etc), we expect a higher speedup for **both** the baseline **and** our method, but the relative improvement will remain similar. This is because we specifically focus on the algorithmic improvement, which is **orthogonal to** the improvements brought by a stronger baseline.
> > >
> > > Given that our reply solved most of your concerns, would you kindly consider increasing your score? We sincerely appreciate your additional time and effort!

---

> > > > ### Author Response · Authors · 2024-12-01
> > > >
> > > > Dear reviewer,
> > > >
> > > > Thank you for investing time and effort in reviewing our paper! We hope you had a wonderful Thanksgiving holiday.
> > > >
> > > > As the author-reviewer discussion period will be closed soon, we would like to check in with you for any additional unresolved questions and concerns. We are happy to discuss it if there is any. Given that both Reviewer 5xbQ and Reviewer KEJf have decided to increase the score, would you like to take a look at the other two reviews and consider increasing your score as well?
> > > >
> > > > We sincerely appreciate your additional time and effort!

---

### Official Review · Reviewer_KEJf · 2024-11-04

**Soundness:** 3
**Presentation:** 3
**Contribution:** 2
**Rating:** 6
**Confidence:** 4

**Summary:**

the paper proposed learning based method for generating adaptive candidate lengths for Speculative Decoding (SpD) algorithm.  SpD algorithm is used for lossless inference acceleration by using a small draft model (run autoregressively) and large target model (run in parallel). The candidate length is usually fixed during SpD run, however, this candidate length affects the speed-up observed with SpD. A fixed candidate length can harm the speed-up depending on the downstream task and the size of the draft model leading to wasteful candidate token generation. The authors train an additional head for predicting when to stop the draft model from generation.

**Strengths:**

The paper solves an existing problem of draft token generation through a learning based method which is interesting as opposed to non-learning based heuristics or hyper-parameter search of fixed draft length.

The motivation for an adaptive draft length is explained well shown with an ideal case example, with proper definitions to discard rate and verification rate.

The training of the additional draft head for predicting draft token generation stop or continue is well explained.

**Weaknesses:**

Though the paper solves an important problem by a learning-based approach the overall gains seem small (7% - 11%) over vanilla speculative decoding.

The paper has not compared with existing heuristic based adaptive draft length methods which use either draft entropy or other confidence scores for stopping or continuing the draft generation

The hyper-parameter search required for finding the w_{reject} and probability threshold is an additional set of hassle also present in vanilla SpD, in this regard it seems like it could be better to simply try vanilla SpD with multiple draft lengths on a small subset of downstream task and then choose one with best performance for entirety of the task.

Lack of comparison with different draft models (TinyLLama-1B) and other models available like OPT-family

Only a single draft model of size 7B was tried, it would be interesting to see what happens to this method with TinyLLama (1B Llama model) and other small llama models.

**Questions:**

Can authors compare their method with non-learning based adaptive draft length approaches?

Can authors compare with other draft models and/or draft-target model families?

Can same adaptive candidate-length draft head be used if the hidden dimension is same for different models or do we need to retrain the adaptive candidate-length draft head? (is the newly trained draft head transferable to different model?)

Extra: why are diffusion models mentioned in the Appendix section B without any relation to adaptive draft length?

Also do the authors have any comments on the future work or what other research problem their work opens?

---

> ### Author Response · Authors · 2024-11-21
>
> We sincerely appreciate the time and effort you have invested in reviewing our paper and providing valuable feedback. We would like to provide detailed responses below:
>
> -----
>
> > **W1**: Though the paper solves an important problem by a learning-based approach the overall gains seem small (7% - 11%) over vanilla speculative decoding.
>
> We would like to re-emphasize the practical contribution of our method:
>
> - (1) The speedup is achieved through **algorithmic improvements** as opposed to system-level improvements. This means that our method can be plugged in whenever the inference method adopts a form of draft models (e.g., early-exiting).  Furthermore, it can be readily combined with other system-level and hardware-level improvements, e.g., fused cuda kernels, smarter KV cache management, or simply faster chips.
> - (2) Our experimental results are **hardware-independent**: Figure 4 shows that our proposed method has better Pareto front than the baseline SpecDec method, which means SpecDec++ will still out-perform the baseline SpecDec under other hardware configurations.   One may argue that the key bottlenecks of speculative sampling is the system-level optimization tricks. Nevertheless, when those bottlenecks are solved, SpecDec++ will still achieve improvement over baseline SpecDec.
> - (3) Besides, achieving an additional 7-11% speed improvement will save 7%-11% of the GPU time and the **financial cost** for those entities that need to serve their LLMs, which could be a huge number for LLM applications with high traffic.
>
> ----
>
> > **W2**: The paper has not compared with existing heuristic based adaptive draft length methods which use either draft entropy or other confidence scores for stopping or continuing the draft generation
>
> **A2**:
>
> We indeed tried using entropy as an indicator during our early investigations but ruled it out as the performance was limited. The specific reasons are:
>
> - (1) Conceptually, the entropy of the predicted token combines two uncertainties: (a) the uncertainty of the next token that is inherent in the natural languages, e.g. the words “good” and “great” have similar meanings and can often be used interchangeably.  (b) the uncertainty related to the confidence of the model’s prediction.  We don’t want to stop the speculation when only (a) is high.
> - (2) According to the speculative decoding algorithm, when the draft model and the target model align very well, the acceptance probabilities are close to 1, no matter whether the current token has a high entropy or not.
> - (3) In the extreme case when the draft model and the target model always give the same distribution, the optimal strategy is to always use the cheaper draft model till the end. However, using entropy of the draft distribution will likely to stop the speculation when it encounters the context where the next token has a high inherent entropy.
>
> To address the problem and distinguish the confidence of the draft model out of the inherent entropy of the natural languages, we work out the theory behind it and train a prediction head to directly predict the acceptance probabilities.
>
>
> ----
>
> > **W3**: The hyper-parameter search required for finding the w_{reject} and probability threshold is an additional set of hassle also present in vanilla SpD, in this regard it seems like it could be better to simply try vanilla SpD with multiple draft lengths on a small subset of downstream task and then choose one with best performance for entirety of the task.
>
> **A3**:  Thank you for pointing out the issue of hyperparameter searches. Please note that **we indeed performed a search over the draft lengths for vanilla SpD**, and showed that our method **even achieves improvement** over **the best** fixed draft length for vanilla SpD:
> - (1) In Figure 1. the reported speed-ups for vanilla SpD (termed SpecDec in our paper) are the best performance for the task over different draft lengths. Our proposed SpecDec++ can achieve performance improvement over the best vanilla SpD.
> - (2) In Figure 4, the Pareto front of the vanilla SpD (termed SpecDec in our paper) is obtained by trying multiple draft lengths. You can see that our proposed SpecDec++ has better Pareto fronts.
>
> About the additionally introduced hyperparameters:
> - (1) We found that the performance of our methods is rather **insensitive** to the parameters $w_{rej}$ and the threshold. Specifically, for $w_rej \in [3, 6]$, setting the threshold from 0.5 to 0.9 achieves nearly the same performance (See the full results in Table 3).
> - (2) Besides, although the inference speeds are different, they **all outperform the baseline SpD method** with a static $K$.
>
> In summary, naively tuning the draft lengths for vanilla SpD has limited performance, and our work utilizes a learned acceptance prediction head to adaptively determine the draft length based on the current prefix, achieving **further speedup** compared to the **optimally tuned** vanilla SpD method.

---

> > ### Author Response · Authors · 2024-11-21
> >
> > > **W4** Lack of comparison with different draft models (TinyLLama-1B) and other models available like OPT-family. Only a single draft model of size 7B was tried, it would be interesting to see what happens to this method with TinyLLama (1B Llama model) and other small llama models.
> >
> > **A4**: Thank you for pointing out the issue.  We repeat all our experiments on a new pair of models: Gemma-1.1-2B-it and Gemma-2-27B-it. We are also able to show performance gains of our proposed SpecDec++ compared to the baseline SpecDec. Please see the [general response](https://openreview.net/forum?id=NnExMNiTHw&noteId=xIyrvcM1Nj) for details!
> >
> > ----
> >
> > ### Questions
> >
> >
> > > **Q1**: Can authors compare their method with non-learning based adaptive draft length approaches?
> >
> > **A1**: Yes. Please see **A2** to **W2** above.
> >
> > ----
> >
> > > **Q2**: Can authors compare with other draft models and/or draft-target model families?
> >
> > **A2**: Yes. Please see **A4** to **W4** above.
> >
> > ----
> >
> > > **Q3**: Can same adaptive candidate-length draft head be used if the hidden dimension is same for different models or do we need to retrain the adaptive candidate-length draft head? (is the newly trained draft head transferable to different model?)
> >
> > **A3**: This is a great question to contemplate. We would expect a performance drop of the prediction head when transferred to a different pair of models.
> >
> > ----
> >
> > > **Q4**: Extra: why are diffusion models mentioned in the Appendix section B without any relation to adaptive draft length? Also do the authors have any comments on the future work or what other research problem their work opens?
> >
> > **A4**: The related work section aims to provide a comprehensive review on reducing the inference latency of language models. We briefly mentioned diffusion language models as the generation time of the diffusion language models can scale sub-linearly with the sequence length, and we believe they have the potential to speed up auto-regressive generation (e.g. as a draft model).
> >
> > Besides, the related work section also mentioned “token tree generation, verification, and pruning”, where our acceptance prediction head can be used as a starting point for pruning the token trees. We believe this is a promising future work direction.
> >
> > -----
> >
> >
> > We sincerely hope that our responses can address your concerns, and we would greatly appreciate it if you would like to re-evaluate our work given the responses.

---

> > > ### Comment · Reviewer_KEJf · 2024-11-25
> > > **response to authors comments**
> > >
> > > Thanks for the new experiment results.
> > > I still feel if this method is transferable or generalizable to different evaluation tasks then it will be a more powerful result and more useful, also if authors can show that training on certain datasets makes the method robust to other datasets.
> > >
> > > I have modified my ratings given the new experiments and author's responses

---

> > > > ### Author Response · Authors · 2024-11-25
> > > > **Thank you for your positive review!**
> > > >
> > > > We greatly appreciate your time and efforts in re-evaluating our paper. For the transferability/generalisability to different tasks, we would like to remind you that our prediction heads were trained only on the Alpaca dataset, so both GSM8K and HumanEval are the out-of-distribution settings. Our experiments indeed showed that the prediction heads can be transferred to different evaluation tasks and even the hyperparameters are robust to the OOD datasets.
> > > >
> > > > Thank you again for your constructive feedback!

---

### Official Review · Reviewer_5xbQ · 2024-11-06

**Soundness:** 3
**Presentation:** 3
**Contribution:** 2
**Rating:** 5
**Confidence:** 4

**Summary:**

This paper introduced an enhancement of speculative decoding by predicting the candidate length on the fly. The algorithm is inspired by a framework that optimizing a corresponding Markov decision process, where an small neural network is trained to predict an optimal threshold. In the experiments, it shows around 10% improvement over vanilla speculative decoding on llama-2 chat 7B& 70B pair.

**Strengths:**

**Theoretical Soundness:** The paper is theoretically grounded, providing a solid foundation for the proposed approach.

**Ease of Implementation:** The method is straightforward to implement, which adds to its practical appeal and accessibility.

**Weaknesses:**

**Limited Practical Improvement:** The reported improvement over vanilla speculative decoding appears minimal, with only a 10% speedup for a large model. Moreover, vanilla speculative decoding is significantly slower than state-of-the-art methods like Medusa [1] and Eagle [2], raising questions about the method's practicality, especially given the additional complexity of training a candidate length prediction model. In terms of theoretical analysis, while the optimal strategy is presented as a threshold function, the threshold’s dependency on both target and source models makes it unclear how a fundamentally residual model can capture this relationship, as the input only includes the context.

**Insufficient Experimentation:** The experiments are limited to a single pair of models and model sizes, which leaves the method’s generalizability unaddressed. It remains unclear if the speedup improvement would increase, decrease, or remain stable with varying model sizes, making it difficult to assess its applicability across different model configurations.

**Lack of Comparisons with State-of-the-Art:** The paper only compares the method against baseline speculative decoding without benchmarking it against more advanced methods. It would be valuable to know if this method could be combined with other speculative decoding techniques and how it performs relative to them.

[1] Cai, Tianle, et al. "Medusa: Simple llm inference acceleration framework with multiple decoding heads." arXiv preprint arXiv:2401.10774 (2024).

[2] Li, Yuhui, et al. "Eagle: Speculative sampling requires rethinking feature uncertainty." arXiv preprint arXiv:2401.15077 (2024).

**Questions:**

- **Training and Parameter Overhead:** How long does it take to train the additional candidate length prediction model? Understanding the time and resource requirements for training this predictor would clarify the overall cost-benefit of the method.

- **Transferability and Generalizability:** Is the predictor transferable or generalizable to other model pairs, or does it require retraining for each new model configuration? Insights on its adaptability would help assess its practicality across different use cases.

- **Performance with Varied Model Pairs:** What is the performance of the method on other model pairs with varying sizes? Testing across different model sizes and configurations would provide a clearer picture of its scalability and effectiveness.

- **Comparison with Other Speculative Decoding Strategies:** How does this method perform relative to other state-of-the-art speculative decoding strategies? It would be valuable to see comparisons with existing methods to better understand its competitive edge.

---

> ### Author Response · Authors · 2024-11-21
>
> We sincerely appreciate the time and effort you have invested in reviewing our paper and providing valuable feedback. We would like to provide detailed responses below:
>
> ----
>
> > **W1**: The reported improvement over vanilla speculative decoding appears minimal, with only a 10% speedup for a large model. Moreover, vanilla speculative decoding is significantly slower than state-of-the-art methods like Medusa [1] and Eagle [2], raising questions about the method's practicality.
>
> **A1**:
> We would like to re-emphasize the contribution of our method:
>
> - (1) The speedup is achieved through **algorithmic improvements** as opposed to system-level improvements. This means that our method can be plugged in whenever the inference method adopts a form of draft models (e.g., early-exiting).  Furthermore, it can be readily combined with other system-level and hardware-level improvements, e.g., fused cuda kernels, smarter KV cache management, or simply faster chips.
> - (2) Our experimental results are **hardware-independent**: Figure 4 shows that our proposed method has better Pareto front than the baseline SpecDec method, which means SpecDec++ will still out-perform the baseline SpecDec under other hardware configurations.   One may argue that the key bottlenecks of speculative sampling is the system-level optimization tricks. Nevertheless, when those bottlenecks are solved, SpecDec++ will still achieve improvement over baseline SpecDec.
> - (3) Besides, achieving an additional 7-11% speed improvement will save 7%-11% of the GPU time and the **financial cost** for those entities that need to serve their LLMs, which could be a huge number for LLM applications with high traffic.
>
> ----
>
> > **W2**: the additional complexity of training a candidate length prediction model.
>
> **A2**:
> The training cost for the prediction head is **marginal** as we only tune a scalar output head **on top of** the last layer of hidden features of the draft model.
> - (1) As a comparison, Medusa [1] involves tuning multiple token prediction heads and the total network size of Medusa heads is larger than ours.
> - (2) In Appendix C.1, we have a rough estimate of the compute resources used in the paper. Training the acceptance head only takes **16.7%** of the total compute. In the real-world setting where we host the model for inference service, the one-time training cost will be essentially negligible compared to the vast amount of the **inference cost**.
>
> ----
>
> > **W3**: In terms of theoretical analysis, while the optimal strategy is presented as a threshold function, the threshold’s dependency on both target and source models makes it unclear how a fundamentally residual model can capture this relationship, as the input only includes the context.
>
> **A3**: Yes the acceptance probability of the current token will depend on both the target model and the draft model. The prediction head is built on top of the draft model and it only includes the context of the previous tokens.
>
> However, such an acceptance prediction head is still able to capture the token acceptance probabilities, because:
> - (1) the **training dataset** is generated using **both** the target model **and** the draft model. Especially, the **ground-truth acceptance probabilities** are computed by calculating the log probability ratios of **the target model** and **the draft model**. Therefore, the training dataset contains the information of the target distribution.
> - (2) We train the acceptance prediction head so it can predict the ground-truth acceptance probabilities. Therefore, the knowledge of the acceptance probabilities of the target model is **trained** into the acceptance prediction head.
> - (3) Besides the above (abstract) theoretical argument, the **effectiveness** of the prediction head model in capturing the relationship between the target model and draft model is **demonstrated by our empirical experiments**.
>
> ----
>
> > **W4**: Insufficient Experimentation: The experiments are limited to a single pair of models and model sizes, which leaves the method’s generalizability unaddressed. It remains unclear if the speedup improvement would increase, decrease, or remain stable with varying model sizes, making it difficult to assess its applicability across different model configurations.
>
> **A4**: We repeat all our experiments on a new pair of models: Gemma-1.1-2B-it and Gemma-2-27B-it. We are also able to show performance gains of our proposed SpecDec++ compared to the baseline SpecDec. Please see the [general response](https://openreview.net/forum?id=NnExMNiTHw&noteId=xIyrvcM1Nj) for details!

---

> > ### Author Response · Authors · 2024-11-21
> >
> > > **W5** Lack of Comparisons with State-of-the-Art: The paper only compares the method against baseline speculative decoding without benchmarking it against more advanced methods. It would be valuable to know if this method could be combined with other speculative decoding techniques and how it performs relative to them.
> >
> > **A5**:
> >
> > - (1) It would be unfair to compare against many SoTA methods in terms of **wall-clock speedups**, as they are typically achieved through combinations of several tricks, e.g., cascading speculative decoding + heavily quantized draft models + other system-level optimizations.
> > - (2) As in **A1**: Our speedup is achieved through **algorithmic improvements** as opposed to other speculative decoding improvements, where our improvement comes from adaptively controlling the draft lengths. Our method can be plugged in with other speculative decoding improvements like EAGLE [2].
> >
> > ----
> >
> > ### Questions
> >
> > > **Q1**: Training and Parameter Overhead: How long does it take to train the additional candidate length prediction model? Understanding the time and resource requirements for training this predictor would clarify the overall cost-benefit of the method.
> >
> > **A1**:  Training the acceptance head only takes 16.7% (500 A100 GPU-hours) of the total compute resources of this paper. In the real-world setting where we host the model for inference service, the one-time training cost will be essentially negligible compared to the vast amount of the **inference cost**.
> >
> > ----
> >
> > > **Q2**: Transferability and Generalizability: Is the predictor transferable or generalizable to other model pairs, or does it require retraining for each new model configuration? Insights on its adaptability would help assess its practicality across different use cases.
> >
> > **A2**: We expect there will be a performance drop when the predictor is transferred to another compatible model pair. To ensure the performance, we recommend retraining the prediction head.
> >
> > ----
> >
> > > **Q3** Performance with Varied Model Pairs: What is the performance of the method on other model pairs with varying sizes? Testing across different model sizes and configurations would provide a clearer picture of its scalability and effectiveness.
> >
> > **A3**: We repeat all our experiments on a new pair of models: Gemma-1.1-2B-it and Gemma-2-27B-it. We are also able to show performance gains of our proposed SpecDec++ compared to the baseline SpecDec. Please see the [general response](https://openreview.net/forum?id=NnExMNiTHw&noteId=xIyrvcM1Nj) for details.
> >
> > ----
> >
> > > **Q4**: Comparison with Other Speculative Decoding Strategies: How does this method perform relative to other state-of-the-art speculative decoding strategies? It would be valuable to see comparisons with existing methods to better understand its competitive edge.
> >
> > **A4**: Please see the answer to **W1**.
> >
> > -----
> >
> > We sincerely hope that our responses can address your concerns, and we would greatly appreciate it if you would like to re-evaluate our work given the responses.

---

> > > ### Comment · Reviewer_5xbQ · 2024-11-21
> > > **Thank you for the response**
> > >
> > > Regarding comparisons with SOTA methods, from my understanding, not all rely primarily on engineering tricks. For instance, Medusa achieves better acceleration by employing multiple speculative sequences in parallel. Does your method support batch speculation? Furthermore, could you provide experimental evidence on how Speculative++ could achieve competitive speedup through certain engineering optimizations?

---

> > > > ### Author Response · Authors · 2024-11-21
> > > >
> > > > Dear reviewer,
> > > >
> > > > Thank you for your quick response.  We would like to provide further responses below:
> > > >
> > > > ### 1. Why did we not compare with or integrate Medusa:
> > > >
> > > > With respect, Medusa does not fall into the category of speculative decoding (as defined in Sec 2 Algorithm 1), as there is no draft model or rejection sampling scheme for importance sampling from the draft distribution. Instead, Medusa uses multiple decoding heads on the last-layer hidden feature of the current token $x_k$ to predict future tokens $(x_{k+1}, \dots, x_{k+M})$ **independently** rather than **auto-regressively**. In other words, the draft distribution of $x_{k+M}$ only depends on $(x_1, \dots, x_{k})$ but not $(x_{k+1}, \dots, x_{k+M-1})$. Therefore, the modeling capacity of the draft token distributions of Medusa heads is limited. For example, the officially released Medusa heads only use **5** heads [(see the huggingface link here)](https://huggingface.co/FasterDecoding/medusa-1.0-vicuna-13b-v1.5/blob/main/config.json) and only supports maximum draft length = 5. In comparison, Speculative decoding supports a much longer draft length.
> > > >
> > > > Medusa is definitely a popular and effective method for speeding up inference. Rigorously speaking, however, Medusa does not fall into the category of speculative decoding, and our adaptive candidate length method cannot be integrated with Medusa, as is discussed in the related work section (Lines 467-469). Many papers aiming at improving speculative decoding algorithms also chose not to compare with Medusa, e.g., DistillSpec [1].
> > > >
> > > > [1] Zhou et al., DistillSpec: Improving Speculative Decoding via Knowledge Distillation https://openreview.net/forum?id=rsY6J3ZaTF
> > > >
> > > > ### 2. Does SpecDec++ support batch speculation?
> > > >
> > > > Yes. SpecDec++ can be extended to batched settings. For the current prefix, we can sample multiple paths of draft tokens independently and apply the acceptance prediction head to determine the length of each path. The downside of this approach is that the supported batch size (of different prompts) will be reduced and it may reduce the overall throughput of the system: e.g. for the single batch setting, the original supported batch size is, say, 256. If we use batch speculation (multiple draft paths) for a single prompt, the supported batch size may be reduced to 128.
> > > >
> > > > ### 3.  Could you provide experimental evidence on how Speculative++ could achieve competitive speedup through certain engineering optimizations?
> > > >
> > > > We would like to emphasize that our method already achieves an absolute competitive speed-up (**2.04x - 2.06x**) through many engineering optimizations (e.g. loading models in half-precisions to reduce I/O overhead, placing the draft model in one GPU to reduce cross-GPU communication cost, ...).
> > > >
> > > > We believe the contribution of the paper is the **relative improvements** over the baseline SpecDec algorithm through adaptive candidate length. Most of the engineering tricks equally apply to the baseline SpecDec method, so having more tricks (e.g. using a 4-bit quantized draft model) will improve the speedups of **both** baseline SpecDec **and** our SpecDec++ without having a significant influence on the **relative improvements**.
> > > >
> > > > Our experimental results focus on the two **hardware-independent** metrics which only depend on the algorithm: discard rate and verification rate. Figure 4 shows our proposed method has better Pareto fronts than the baseline SpecDec method, which means SpecDec++ will still outperform the baseline SpecDec if other tricks are deployed.
> > > >
> > > > ---
> > > >
> > > > Again, thank you for the time and effort you have spent reviewing our paper. We sincerely hope that our new responses can address your concerns, and we would greatly appreciate it if you would like to re-evaluate the academic value of our work.

---

> > > > > ### Comment · Reviewer_5xbQ · 2024-11-22
> > > > >
> > > > > Thank you for your response. While the methodology presented in this work appears sound, I remain concerned about its practical applicability due to the marginal improvement it offers over existing speculative decoding techniques. As another reviewer also pointed out, demonstrating your method in combination with stronger algorithms could have significantly bolstered its relevance and utility in practice. Without this, the contributions appear limited.
> > > > >
> > > > > Furthermore, I would like to clarify my perspective on Medusa's heads. In my view, they can still be considered a form of draft models. Their strength lies in verifying multiple speculations in parallel, leveraging the computational power of GPUs. This design minimizes the requirement for each speculation to be highly accurate, focusing instead on generating a large number of moderately accurate speculations. This approach leads to substantially faster performance compared to traditional speculative decoding.
> > > > >
> > > > > Despite its divergence from the original model's sampling distribution, Medusa preserves identical outputs under top-1 decoding, ensuring no degradation in deterministic tasks. This equivalence to the original model under top-1 decoding further strengthens Medusa’s practicality.
> > > > >
> > > > > Finally, I believe that the limitation of long speculation lengths further diminishes the relevance of the proposed method. In practice, speculative decoding methods, including Medusa, typically exhibit a significant drop in acceptance length after the third token. If this were not the case, state-of-the-art speedups would reach 4x or more. Medusa's design hits an optimal balance with five heads, as predictions from subsequent heads beyond the third often exhibit poor accuracy. While theoretically, more heads can be trained, their practical utility is minimal.

---

> > > > > > ### Author Response · Authors · 2024-11-22
> > > > > > **fundamental misunderstanding of speculative decoding methods**
> > > > > >
> > > > > > Thank you for engaging in the discussion and sharing your view on the field. It appears you have some fundamental misunderstanding of speculative decoding methods and even Medusa-type methods.
> > > > > >
> > > > > > > 1. Medusa's strength lies in verifying multiple speculations in parallel, leveraging the computational power of GPUs. This design minimizes the requirement for each speculation to be highly accurate, focusing instead on generating a large number of moderately accurate speculations.
> > > > > >
> > > > > > Traditional speculative decoding methods also verify multiple speculations in parallel, e.g., the token-tree verification scheme in Sequoia [1], EAGLE [2], and MCSD [3]. Besides, for both traditional speculative decoding and Medusa, improving the accuracy of the speculation and increasing the total number of draft tokens in parallel are two important directions toward further speed-ups. Please note that Medusa-2 (compared to Medusa-1, which only trains the new heads) was proposed by the same authors to **improve the speculative prediction ability** through full-model fine-tuning [(link)](https://github.com/FasterDecoding/Medusa). It is irresponsible to say "Medusa design minimizes the requirement for each speculation to be highly accurate", which discredits the efforts of Medusa-2.
> > > > > >
> > > > > > - [1] Zhuoming Chen et al. Sequoia: Scalable, robust, and hardware-aware speculative decoding.
> > > > > > - [2] Yuhui Li et al . Eagle: Speculative sampling requires
> > > > > > rethinking feature uncertainty.
> > > > > > - [3] Sen Yang et al.  Multi-candidate speculative decoding.
> > > > > >
> > > > > > > 2. Despite its divergence from the original model's sampling distribution, Medusa preserves identical outputs under top-1 decoding, ensuring no degradation in deterministic tasks. This equivalence to the original model under top-1 decoding further strengthens Medusa’s practicality.
> > > > > >
> > > > > > Please note that traditional speculative decoding is **guaranteed** to recover the target model's sampling distribution (see Theorem 1 in [4]), so traditional speculative decoding ensures no degradation in **all** tasks.
> > > > > > If not because of a fundamental misunderstanding of the traditional speculative decoding methods or a subjective bias toward Medusa, then it is extremely irresponsible and misleading to package the weakness of Medusa into strength and undermine all the existing and ongoing efforts of the traditional speculative decoding.
> > > > > >
> > > > > > - [4] Chen et al., Accelerating Large Language Model Decoding with Speculative Sampling
> > > > > >
> > > > > > > 3. Finally, I believe that the limitation of long speculation lengths further diminishes the relevance of the proposed method. In practice, speculative decoding methods, including Medusa, typically exhibit a significant drop in acceptance length after the third token.
> > > > > >
> > > > > > You mention that "speculative decoding methods, including Medusa, typically exhibit a significant drop in acceptance length after the third token", which is an extremely strong claim without evidence. We respectfully point out that **this is a wrong claim**.
> > > > > > - (1) In the corner case when the draft model distribution always matches the target model distribution, the theoretical acceptance length is infinite (see Corollary 3.6. of [5]).
> > > > > > - (2) In the practical setting, if you take a look at our Fig. 4, the traditional speculative decoding can have a verification rate < 0.15 at GSM8K and HumanEval datasets, which corresponds to an average accept length of **6.9 tokens**.
> > > > > >
> > > > > > [5] Leviathan et al., Fast Inference from Transformers via Speculative Decoding
> > > > > >
> > > > > > > 4. If this were not the case, state-of-the-art speedups would reach 4x or more. Medusa's design hits an optimal balance with five heads, as predictions from subsequent heads beyond the third often exhibit poor accuracy. While theoretically, more heads can be trained, their practical utility is minimal.
> > > > > >
> > > > > > We respectfully point out that it is oversimplified to draw an equivalence between the typical acceptance length and the overall speedups. Our paper has dedicated multiple sections to analyze the factors that determine the overall speedup (see Sec 2. Lines 145-175; Sec 4.2). Specifically, the speedup depends on three factors: (1) discard rate, which measures how often the draft tokens are wasted, (2) verification rate, which is the reciprocal of the acceptance length, and (3) the ratio of the forward time of the target model to the draft model. The speedup is determined through Eq. (2.3) in our paper (Line 171) and we have empirical evidence supporting its correctness.
> > > > > >
> > > > > > The reason why predictions from Medusa heads beyond the third often exhibit poor accuracy is: the Medusa heads are **independent** instead of **auto-regressive**. In plain words, think about the accuracy of predicting the fourth subsequent token without specifying what the previous three tokens are.
> > > > > >
> > > > > > -----
> > > > > >
> > > > > > The contributions have been emphasized in our previous replies. If you are still not convinced of the value of the paper, please at least consider **lowering your confidence scores**. Thank you for your time and efforts.

---

> > > > > > > ### Comment · Reviewer_5xbQ · 2024-11-25
> > > > > > >
> > > > > > > Thank you for the response. I brought up Medusa because, in my opinion, multi-speculation in parallel is crucial for improving the speed of speculative decoding.
> > > > > > >
> > > > > > > From your results, it appears that the impact of adaptive candidate length on performance is **not particularly significant**. Integrating your method with multi-speculation might be a promising way to enhance performance. Furthermore, integrating your method with SpecTr [1] seems feasible. However, this integration might further diminish the role of adaptive candidate length, necessitating additional experiments to evaluate this hypothesis.
> > > > > > >
> > > > > > > While I acknowledge your contributions from an algorithmic perspective, it is worth noting that your approach requires **additional model components**, unlike standard speculative decoding (SD). Therefore, it is not strictly a plug-and-play improvement over SD. Moreover, from a purely theoretical perspective, there is no guarantee that the length prediction head can improve performance, as the candidate length prediction head’s outputs are not guaranteed to be accurate.
> > > > > > >
> > > > > > > To make your results more compelling, I recommend conducting **comprehensive experiments across diverse model pairs** (e.g., varying size ratios like a 7B-1B pair) and environments. More empirical experiments are essential to understanding the practical conditions under which your proposed method works effectively.
> > > > > > >
> > > > > > > Another reviewer also suggested considering TinyLlama, so why not explore this option to further complete your study within the Llama family instead of switching to the Gemma pair? Given that the results for Gemma are worse than those for Llama, is Llama the best-performing model in your experiments?
> > > > > > >
> > > > > > > Overall, given the soundness of the methodology, I have decided to increase my score. However, I believe there is still room for improvement in the experiments as described above.
> > > > > > >
> > > > > > > [1] SpecTr: Fast speculative decoding via optimal transport. (2023). *arXiv*. https://arxiv.org/abs/2309.12483

---

> > > > > > > > ### Author Response · Authors · 2024-12-01
> > > > > > > > **Thank you for your additional constructive feedback**
> > > > > > > >
> > > > > > > > Dear reviewer,
> > > > > > > >
> > > > > > > > Thank you for all your detailed responses, the additional constructive feedback, and your willingness to re-evaluate the work and increase the score! We hope you had a wonderful Thanksgiving holiday.
> > > > > > > >
> > > > > > > > We agree with you that integrating our method with token tree methods (multi-speculation) would be a promising way to enhance performance. However, instead of diminishing the role of the acceptance prediction head, the multi-speculation setting must be **built upon** the acceptance prediction head. To extend the **adaptive candidate length** method to the **adaptive token tree** method, we still need a method to adaptively determine which tree leaf node to expand. Intuitively, we want to select the token tree node with a high acceptance probability to expand. To make the intuition rigorous and sound, we agree with you that additional experiments are needed to evaluate this hypothesis and this will be an interesting follow-up work. While our work focuses on the simple linear setting, we believe our results can be a starting point for both the theoretical analysis and the acceptance prediction head training method in the token tree setting.
> > > > > > > >
> > > > > > > > Regarding the additional experiments, we chose to switch to the Gemma family because we prioritized diversifying model families over different model scales within the same family. we will work on the Tinyllama-1.1B-chat v/s Llama2-chat 7B & 70B as  suggested.
> > > > > > > >
> > > > > > > > Thank you again for the time and effort you have spent reviewing our paper!

---

### Meta-Review · Area_Chair_Ba6C · 2024-12-19

**Metareview:**

(a) Summary of Scientific Claims and Findings

This paper introduces SpecDec++, an innovative speculative decoding algorithm that dynamically adjusts the length of candidate tokens. The authors frame candidate length selection as a Markov Decision Process (MDP) and propose a threshold-based stopping policy.

(b) Strengths of the Paper

1. SpecDec++ effectively adapts candidate lengths, resulting in greater speedup.

2. It demonstrates notable speedups over SpecDec across multiple datasets.

(c) Weaknesses of the Paper and Missing Elements

1. The paper lacks comparisons with recent state-of-the-art methods like Medusa and EAGLE.
2. Without such comparisons, it is challenging to conclude that SpecDec++ outperforms Medusa and EAGLE in terms of performance gains.

(d) Decision and Rationale

Additional discussions and comparisons with Medusa and EAGLE are essential to strengthen the paper.

**Additional Comments On Reviewer Discussion:**

The authors have addressed many of the reviewers’ concerns; however, the discussions regarding Medusa and EAGLE fail to fully persuade the reviewers.

---

### Decision · Program_Chairs · 2025-01-22

Reject